# Interfacial self-healing polymer electrolytes for long-cycle solid-state lithium-sulfur batteries

Fei Pei [1], Lin Wu[1], Yi Zhang[1], Yaqi Liao[1], Qi Kang [1], Yan Han[1], Huangwei Zhang[1], Yue Shen [1], Henghui Xu [1] ✉, Zhen Li[1] ✉ & Yunhui Huang [1] ✉

Coupling high-capacity cathode and Li-anode with solid-state electrolyte has been demonstrated as an effective strategy for increasing the energy densities and safety of rechargeable batteries. However, the limited ion conductivity, the large interfacial resistance, and unconstrained Li-dendrite growth hinder the application of solid-state Li-metal batteries. Here, a poly(ether-urethane)-based solid-state polymer electrolyte with self-healing capability is designed to reduce the interfacial resistance and provides a high-performance solid-state Li-metal battery. With its dynamic covalent disulfide bonds and hydrogen bonds, the proposed solid-state polymer electrolyte exhibits excellent interfacial self-healing ability and maintains good interfacial contact. Full cells are assembled with the two integrated electrodes/electrolytes. As a result, the Li∥Li symmetric cells exhibit stable long-term cycling for more than 6000 h, and the solid-state Li-S battery shows a prolonged cycling life of 700 cycles at 0.3 C. The use of ultrasound imaging technology shows that the interfacial contact of the integrated structure is much better than those of traditional laminated structure. This work provides an interesting interfacial dual-integrated strategy for designing high-performance solid-state Li-metal batteries.

Solid-state Li-metal batteries (SSLMBs) have received widespread attention due to their high energy densities and safety[1–3]. With a Li-metal anode and a sulfur-based cathode, the energy density of the full cell is greatly enhanced[4,5]. However, the complex and unstable interfacial problems of the electrolytes and electrodes, such as continuous polysulfide shuttling[6,7], rapid consumption of the electrolyte and the uncontrollable growth of Li-metal dendrites[8–12], have limited commercialization of the lithium-sulfur (Li-S) batteries[13,14].

The use of solid-state electrolytes is an effective way to solve the inherent safety problems of Li-metal batteries[15–17]. Compared with inorganic solid-state electrolytes (e.g., oxide- and sulfide-based solid electrolytes), which have high ionic conductivities and thermal stabilities but are limited by severe mechanical brittleness and high electrode/electrolyte interfacial resistance[3,18,19], solid-state polymer electrolytes (SPEs) show excellent mechanical flexibility and electrochemical stability[20–22]. As representative SPEs, poly(ethylene oxide) (PEO)-based electrolytes are widely studied because of their excellent lithium salt solvation capacities[23]. However, the poor Li-ion conductivity caused by the high crystallinity, poor mechanical strength, and unsatisfactory electrochemical stability has restricted further application[24–26]. In fact, it is an exaggeration to imagine that the flexibility of an all-solid-state polymer electrolyte would solve the interfacial contact problem of SSLMs[24,27]. With the interfacial ion transfer barrier between the solid-state polymer electrolyte and the uneven electrode, the true ionic conductivity of the polymer electrolyte cannot be fully utilized[28]. In situ polymerization is an effective way to solve the interface problems of SPEs[29,30]. However, in situ polymerization reactions are limited to a few monomer types, and the degrees of polymerization are uncontrollable, leading to gels or semisolid final products[31,32]. The rational design of self-healing SPEs

[1]State Key Laboratory of Materials Processing and Die & Mould Technology, School of Materials Science and Engineering, Huazhong University of Science and Technology, Wuhan 430074, China. ✉e-mail: xuhh@hust.edu.cn; li_zhen@hust.edu.cn; huangyh@hust.edu.cn

with using disulfide bonds or hydrogen bonds is intended to inhibit cracks or deformation of the electrode during cycling[33–36]. However, the assembled battery is a black box, and there are no effective ways to track the solid–solid-interfacial contact problems in real time. The need to enhance the ionic conductivity and construct efficient interfacial ion transport paths through molecular structures remains serious challenges in the development of SSLMBs.

Herein, we propose a new class of poly(ether-urethane)-based SPEs for constructing integrated SSLMBs and achieving excellent electrochemical performance. The abundant ether-oxygen and carbonyl functional groups in the structure enable dissociation of the lithium salts and improve the ionic conductivity. More importantly, over the whole life cycle, the dynamic covalent disulfide bonds and the hydrogen bonds between urethane groups provide excellent interfacial self-healing ability to repair solid/solid interfacial defects. In situ ultrasound imaging results shows that the interfacial contact of the integrated solid-state electrode/electrolyte structure is much better than those of traditional laminated structures. As a result, the enhanced interfacial contact enables the sulfurized polyacrylonitrile (SPAN) cathode to deliver a significantly enhanced cycling stability (93% capacity retention after 700 cycles at 0.3 C) and rate performance (560 mAh $g_{SPAN}^{-1}$ at 1 C). When matched with a sulfur/carbon black (S@CB) cathode, the solid-state Li-S battery was cycled over 350 times with a high capacity of 812 mAh $g_S^{-1}$.

## Results

### Structural design of SPEs and integrated SSLMBs

Poly(ether-urethane) polymers are usually formed by stepwise polymerization of polyalcohols and polyisocyanates, and the urethane repeat group (-NH-COO-) is formed in the main framework[37,38]. Selection of monomers with appropriate molecular weights is very important, and the activity of the terminal hydroxyl is affected by the lengths of the molecular chains (Supplementary Fig. 1). Polytetrahydrofuran (PTMG) with a low molecular weight (Mw=2,000 g mol⁻¹) and ether-oxygens was chosen as the soft segment to dissociate and conduct the lithium ions (Li⁺)[39]. The terminal hydroxyls of PTMG reacted readily with hexamethylene diisocyanate (HDI) to generate -NH-COO- groups (Fig. 1a). On this basis, the poly(ether-urethane) with self-healing character (PTMG-HDI-BHDS) was synthesized by introducing 2-hydroxyethyl disulfide (BHDS) as a chain extender, and the hydroxyl groups of BHDS were polymerized with excess -N = C = O. The repeated reaction nodes -NH-COO- in the polymer chain formed a rich hydrogen bond network[40,41]. The reaction conditions were mild and very easily scaled up to the kilogram level (Supplementary Fig. 2). Moreover, by introducing the dynamic disulfide bonds with low bond dissociation energy into the poly(ether-urethane), rearrangement of the disulfide bonds occurs readily at low temperatures and without external stimulation to enable rapid self-healing of PTMG-HDI-BHDS at room temperature (Fig. 1b)[42–45].

To show the two completely different battery preparation strategies more vividly, the structures of the laminated and integrated batteries were compared. With the laminated assembly method, the traditional SSLMB always shows high interfacial resistance due to the uneven interface and the presence of pore gaps (Fig. 1c), which have plagued SSLMBs for a long time[46]. The dual-integrated SSLMB was constructed with an interfacial self-healing SPE (Fig. 1d), and the integrated sulfur cathode (S@SPE) and integrated anode (Li@SPE) were prepared by preimpregnation coating. During solvent volatilization

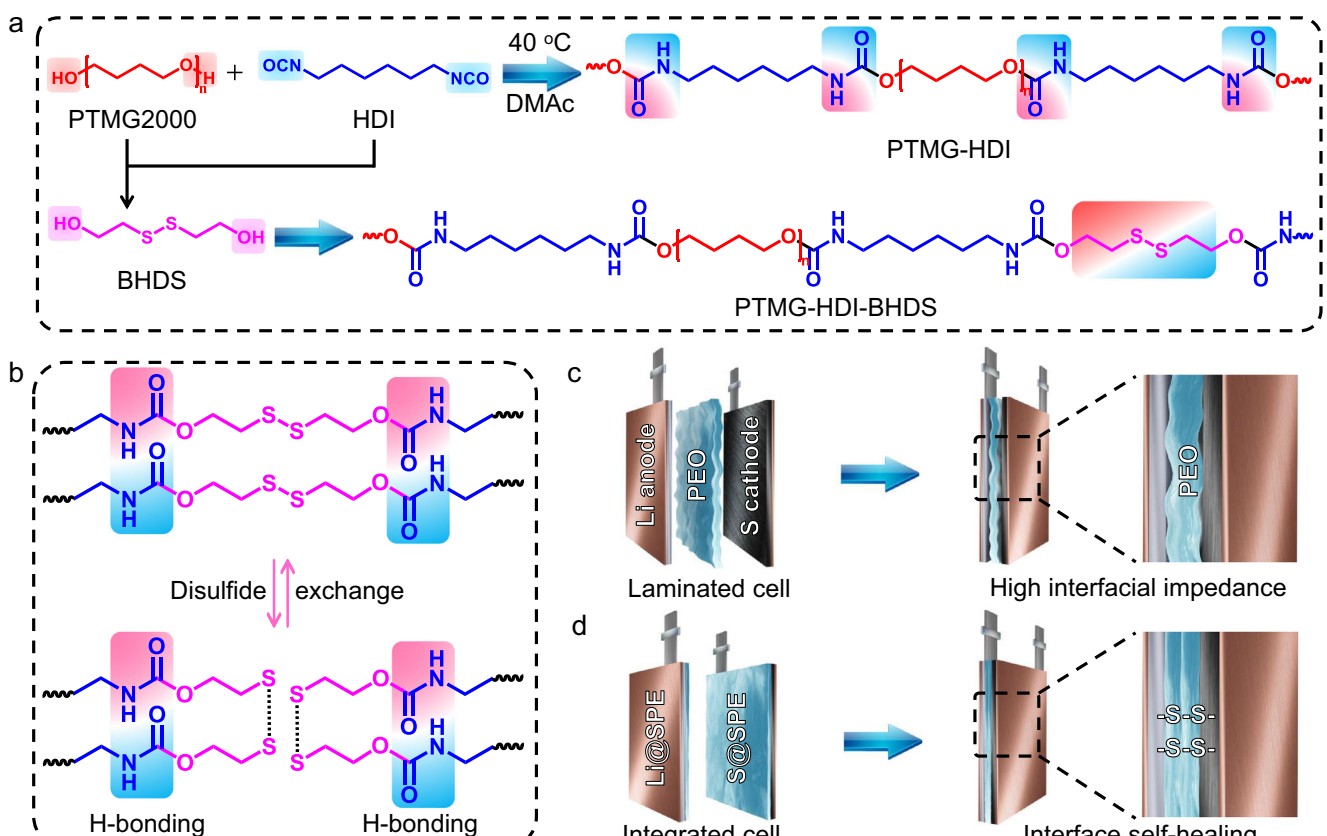

**Fig. 1 | Schematic illustration of the polymer electrolyte structure and structural comparison of solid-state Li-S batteries. a** Chemical structures of PTMG-HDI and PTMG-HDI-BHDS. **b** Schematic structures of the dynamic covalent disulfide bonds and hydrogen bonds in PTMG-HDI-BHDS. **c** Structure of a conventional laminated solid-state Li-S battery. **d** Structure of the integrated electrode/electrolyte and solid-state Li-S battery.

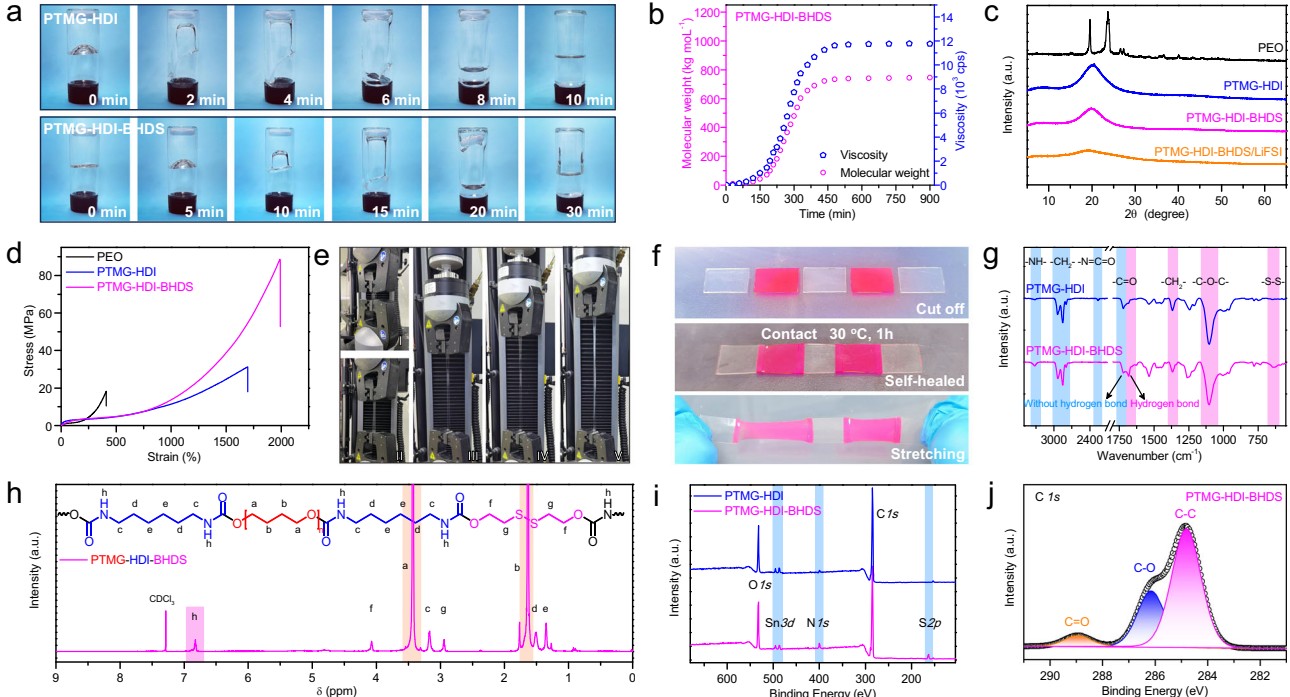

**Fig. 2 | Structural characterization of the polymers. a** Viscosity contrast optical photographs of PTMG-HDI and PTMG-HDI-BHDS after polymerization. **b** Variations in molecular weight (red point) and viscosity (blue point) during the polymerization process. **c** XRD patterns of the polymers and PTMG-HDI-BHDS/LiFSI. **d** Stress– strain curves for the polymers. **e** Photographs of the stress–strain measurements of the PTMG-HDI-BHDS. **f** The interfacial self-healing process of PTMG-HDI-BHDS. **g** ATR-FTIR spectra of the polymers. **h** $^1$H NMR spectrum of PTMG-HDI-BHDS. **i** Comparison of the XPS survey spectra. **j** C *1s* XPS data for PTMG-HDI-BHDS.

and solidification, the electrode/electrolyte interface was welded together, and finally, the role of disulfide bonds with self-healing character mainly focused on solving the interface contact problem between the SPE layers on the surface of the two integrated electrodes.

## Material fabrication and characterization of the SPEs

Both PTMG-HDI and PTMG-HDI-BHDS solutions exhibited uniform and transparent gel states (Fig. 2a). The difference was that the wall hanging time of PTMG-HDI-BHDS was significantly longer than that of PTMG-HDI, because the molecular weight and viscosity of PTMG-HDI-BHDS were increased by the introduction of a chain extender. The real-time polymerization process of PTMG-HDI-BHDS was monitored, and the average molecular weight ($M_n$) and viscosity ($\eta$) were measured with gel permeation chromatography (GPC) (Fig. 2b). The viscosity increased synchronously with the molecular weight during a reaction time of 8 h at 40 °C, and $M_n$ and $\eta$ ultimately reached $7.3 \times 10^5$ g mol$^{-1}$ and $1.3 \times 10^4$ cps, respectively, which were significantly higher than those of PTMG-HDI. The prepared poly(ether-urethane)-based polymer can be stored for a long time due to its excellent thermal and chemical stability (Supplementary Fig. 3).

The PTMG-HDI-BHDS solution was further cast on a polytetra-fluoroethylene plate, and an optically transparent film was obtained after solvent evaporation (Supplementary Fig. 4). Compared with those of PEO and PTMG-HDI, the scanning electron microscopy (SEM) images showed that the surface of PTMG-HDI-BHDS was very dense and smooth (Supplementary Fig. 5). The crystallinity status was analyzed with XRD (Fig. 2c). PEO presented two strong peaks at approximately 19.5° and 23.6°. After the polymer chains of PTMG-HDI were extended with BHDS, the peak intensities for PTMG-HDI-BHDS and PTMG-HDI-BHDS/LiFSI were obviously reduced, and the decreased crystallinity improved the Li$^+$ mobility and ion conductivity[47].

The breaking strength and ultimate elongation of PTMG-HDI-BHDS were 88.3 MPa and 2000%, respectively, which were significantly higher than those of PEO (17.8 MPa) and PTMG-HDI (31.2 MPa) (Fig. 2d,

2e and Supplementary Fig. 6)[48–50]. This was attributed to the extended hydrogen-bond network and the increased molecular weight caused by the chain extender, indicating that PTMG-HDI-BHDS could theoretically serve as a mechanical barrier to suppress Li dendrites growth[51]. The self-healing ability of PTMG-HDI-BHDS was evaluated with the artificially stained method, and rhodamine B was added to the colorless PTMG-HDI-BHDS solution to obtain a red film (Fig. 2f and Supplementary Fig. 7). The PTMG-HDI-BHDS film segments with different colors completely merged into an integral block, and the self-healing interfaces were strong enough to withstand a tensile force along the direction vertical to the cut surface. Furthermore, the self-healing behavior was also confirmed by analyzing the PTMG-HDI-BHDS film after scratching with a scalpel (Supplementary Fig. 8). As shown in the Fourier transform infrared spectrum (ATR-FTIR) (Fig. 2g), a strong C-O-C stretching vibrational peak appeared at approximately 1102 cm$^{-1}$, and peaks for the urethane groups (-NH stretching vibration at 3325 cm$^{-1}$ and C = O stretching vibration at 1719 cm$^{-1}$) and disulfide bonds (approximately 635 cm$^{-1}$) were clearly observed after polymerization, suggesting successful synthesis of the PTMG-HDI-BHDS. The peak for the C = O groups were split into two peaks, one of which was blue shifted to approximately 1680 cm$^{-1}$ due to the formation of strong hydrogen bonds between the -NH-COO- groups (Fig. 2g and Supplementary Fig. 9). The functional groups of the pristine and self-healed PTMG-HDI-BHDS were well coincident by ATR-FTIR, and the mechanical strength of the self-healed PTMG-HDI-BHDS were very close to the original value (Supplementary Fig. 10).

$^1$H NMR spectra was recorded to determine the structure of the polymer (Fig. 2h and Supplementary Fig. 11). The strong signals at 3.43 and 1.63 ppm were assigned to the H on the repeat unit (-O-CH$_2$-CH$_2$-CH$_2$-CH$_2$-O-) of PTMG. The 1:1:1 ratio for the integrated areas of the peaks at 3.17, 1.51 and 1.35 ppm were consistent with the structure of the HDI repeat unit (Supplementary Fig. 12). The 1:1 ratio for the integrated areas of the peaks at 4.08 and 2.94 ppm in PTMG-HDI-BHDS were consistent with the -S-CH$_2$-CH$_2$-S- units in BHDS (Fig. 2h),

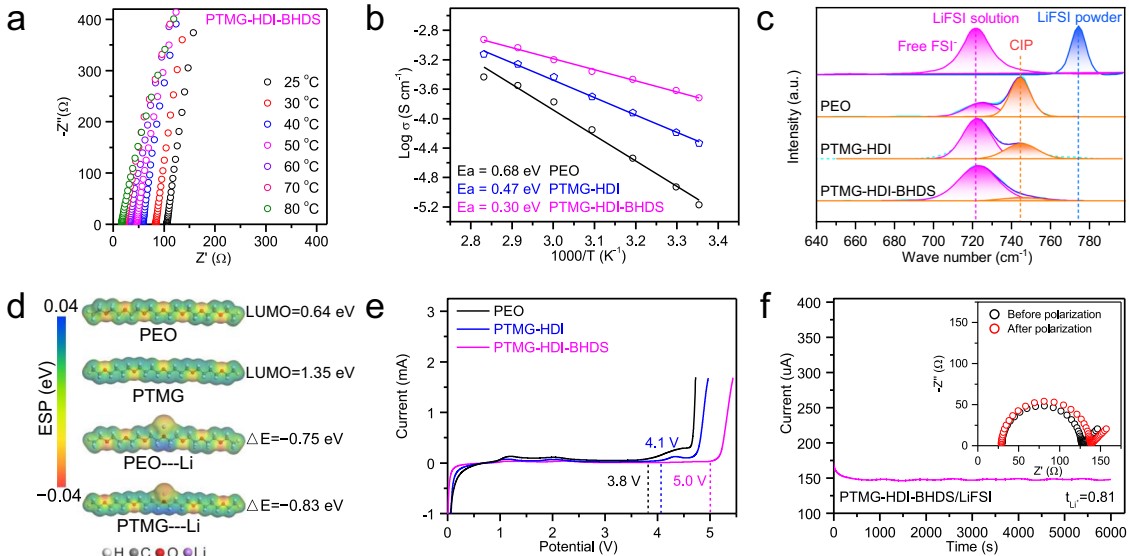

**Fig. 3 | Electrochemical characterization of the SPEs. a** EIS data for PTMG-HDI-BHDS/LiFSI determined at different temperatures. **b** Temperature-dependent ionic conductivities of the SPEs. **c** Raman spectra and the corresponding fitting curves for the S-N-S stretching vibrational mode of the SPEs. **d** Calculated lowest unoccupied molecular orbitals (LUMOs) of PEO and PTMG and their binding energies with Li⁺. **e** Linear scan voltammograms for the SPEs. **f** Chronoamperometry curve and AC impedance spectra before and after polarization of the Li|PTMG-HDI-BHDS|Li symmetric cell.

confirming that the chain extender was successfully introduced into the polymer.

High-resolution C $1s$ and N $1s$ X-ray photoelectron spectroscopy (XPS) and survey spectra were generated (Fig. 2i, 2j and Supplementary Fig. 13). The intensity of the N $1s$ peak for PTMG-HDI-BHDS was significantly higher than that for PTMG-HDI, and a new peak (163.4 eV) which was assigned to the disulfide bonds appeared, indicating that the reaction with the chain extender increased the urethane content. The C $1s$ spectrum (Fig. 2j) was deconvoluted into three different peaks of C-C (284.8 eV), C-O-C (286.2 eV), and C=O (288.9 eV), consistent with the molecular structure of PTMG-HDI-BHDS (Fig. 2h).

## Electrochemical characterization of the SPEs
The Li⁺ conductivities of the SPEs at temperatures ranging from 25 to 80 °C are presented in Fig. 3a and Supplementary Fig. 14. The dual-integrated SS|PTMG-HDI-BHDS|SS (SS stands for sheet steel) cell was assembled by fitting two integrated SS@SPEs together, and the total thickness of the bonded SPE was approximately 200 μm to reduce the measurement error (Supplementary Fig. 15). By adjusting the amount of LiFSI, the optimal weight ratio of polymer:LiFSI=2:1 was established (Supplementary Fig. 16). PTMG-HDI-BHDS/LiFSI ($2.4 \times 10^{-4}$ S cm⁻¹) exhibited a significantly higher ionic conductivity than PTMG-HDI/LiFSI ($6.5 \times 10^{-5}$ S cm⁻¹) and PEO/LiFSI ($1.2 \times 10^{-5}$ S cm⁻¹) at 30 °C, indicating that the strong interfacial self-healing and adhesion significantly decreased the interfacial resistance of the SS@SPE, and the amorphous polymer skeleton enabled fast Li⁺ transport. The ionic conductivities at different temperatures are presented in Fig. 3b. The lowest activation energy was seen in the Arrhenius plot for PTMG-HDI-BHDS/LiFSI (Ea = 0.30 eV), and it reflected a lower Li⁺ migration barrier compared to those of PTMG-HDI/LiFSI and PEO/LiFSI, which improved the rate performance of the battery.

Raman spectra were used to evaluate the dissociation of LiFSI in the SPEs[52]. The S−N−S peaks of FSI⁻ at 721, 744, and 773 cm⁻¹ were represented free FSI⁻ (completely dissolved in water), contact ion pairs (CIP), and undissociated LiFSI, respectively (Fig. 3c). The LiFSI solvation structures in the three SPEs were differentiated by comparing the integrated area of each peak. PEO/LiFSI showed a high proportion of CIPs and very limited free FSI⁻ because of the relatively weak dissociation capacity of PEO. In stark contrast, the S−N−S peak for

PTMG-HDI-BHDS/LiFSI underwent a significant shift to 721 cm⁻¹, and the proportion of free FSI⁻ increased from 68.5% (PTMG-HDI/LiFSI) to 95.2% (PTMG-HDI-BHDS/LiFSI), indicating that the low crystallinity of PTMG-HDI-BHDS facilitated dissociation of the LiFSI. The LUMO energy of the repeat unit in PTMG (1.35 eV) was significantly higher than that of the PEO repeat unit (0.64 eV), indicating stronger electrochemical compatibility of PTMG-HDI-BHDS with the Li anode (Fig. 3d)[53]. The binding energy of PTMG ($\Delta E_{PTMG} = -0.83$ eV) with Li⁺ was higher than that of PEO ($\Delta E_{PEO} = -0.75$ eV), which accelerated the solvation and dissociation of Li⁺ with PTMG-HDI-BHDS.

The linear sweep voltammograms (LSV) for the asymmetrical cells indicated that PTMG-HDI-BHDS/LiFSI displayed a wider electrochemical voltage beyond 5.0 V than PEO (3.8 V) and PTMG-HDI (4.1 V), which is valuable for high-voltage (e.g., LiNi₀.₈Co₀.₁Mn₀.₁O₂) battery systems, the corresponding PEO and PTMG-HDI showed significantly poor compatibility (Fig. 3e and Supplementary Fig. 17). The improved interfacial electrochemical stability of PTMG-HDI-BHDS/LiFSI was attributed to the adjacent electron-donating groups of the urethane, which suppressed decomposition of the adjacent E-O segments. In addition, the exceptionally high Li⁺ transference number ($t_{Li}^+ = 0.81$) for PTMG-HDI-BHDS/LiFSI could inhibit the formation of space charge near the Li anode and reduce dendrite formation at a high current density (Fig. 3f). The abundant urethane groups generated an extended -NH····F hydrogen bond network with the FSI⁻ anions, which hindered the transport of FSI⁻ to a certain extent[54]. This was confirmed by the upfield displacement of the peak for FSI⁻ in the ¹⁹F spectrum (Supplementary Fig. 18).

## Electrochemical stabilities of the Li|SPEs|Li symmetric cells
Ultrasound imaging is a powerful tool with which to evaluate interfacial contacts and side reactions due to its high sensitivity to gas/vacuum[55,56]. The peak-peak amplitude values (PPVs) of the recorded maximum and minimum signals are converted into different colors ranging from blue to red to create the ultrasonic image. Ultrasonic imaging was carried out during galvanostatic cycling of the Li|SPEs|Li pouch cells at 0.2 mA cm⁻² (Fig. 4a–f and Supplementary Fig. 19). It is worth noting that the dual-integrated strategy cannot be adopted with PEO/LiFSI because the significant side reaction between the solvent of acetonitrile and Li (PEO was dissolved in acetonitrile). A large blue area

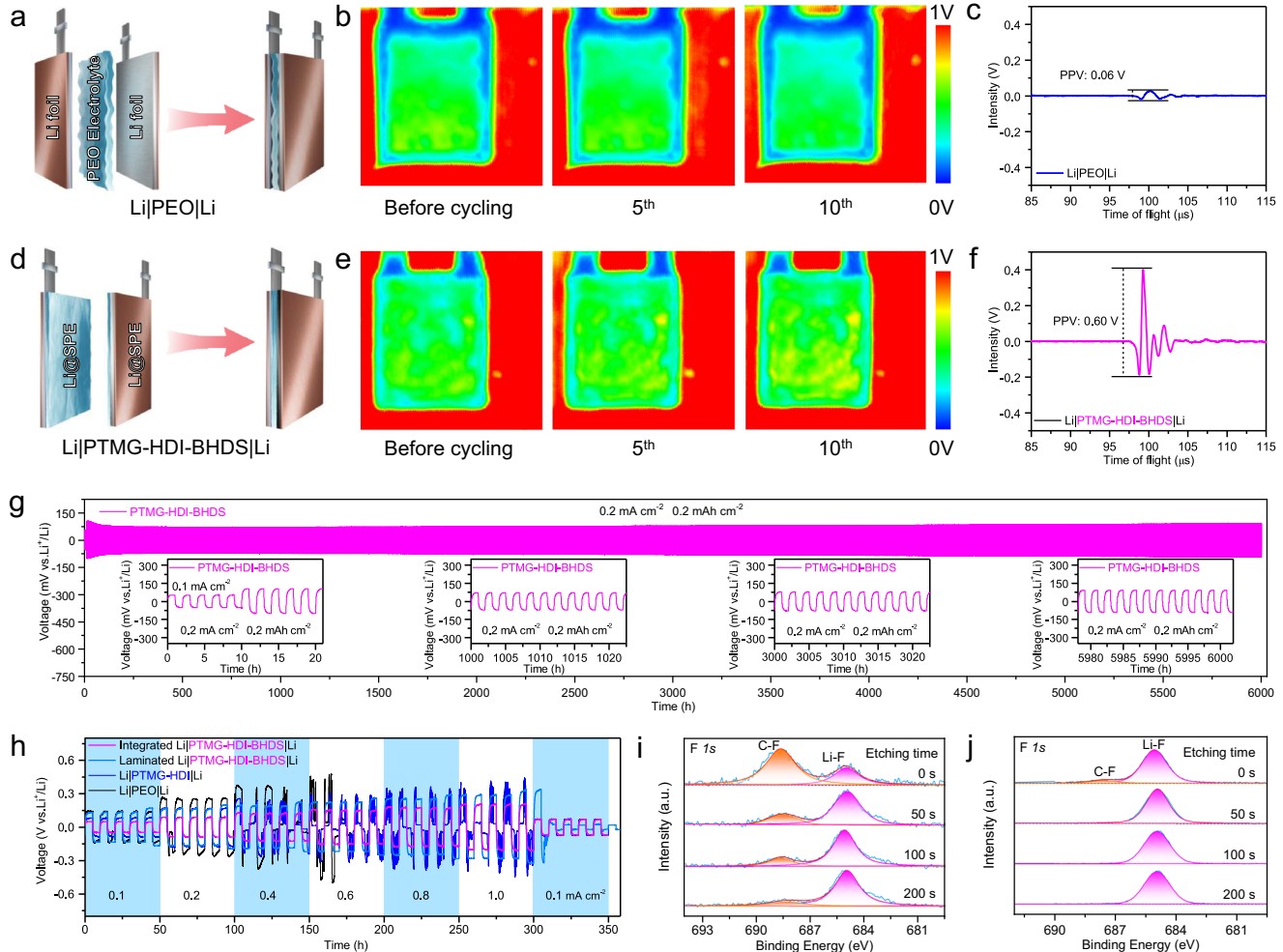

**Fig. 4 | Electrochemical stability of the Li|SPEs|Li symmetric cells. a** Internal structure of Li|PEO|Li. **b, c** Ultrasonic transmission images of a Li|PEO|Li pouch cell during the first 10 cycles (**b**) and the corresponding ultrasonic waves (**c**). **d** The internal structure of Li|PTMG-HDI-BHDS|Li. **e, f** Ultrasonic transmission images of the Li|PTMG-HDI-BHDS|Li pouch cell during the first 10 cycles (**e**) and the corresponding ultrasonic waves (**f**). **g** Galvanostatic cycling of a Li|PTMG-HDI-BHDS|Li coin cell at 0.2 mA cm$^{-2}$. Inset: representative voltage profiles. **h** Rate capabilities of the Li|SPEs|Li symmetric cells. **i, j** Depth-profiled XPS datas for the cycled Li anodes in (**i**) Li|PEO|Li and (**j**) Li|PTMG-HDI-BHDS|Li cells.

appeared around the edges of the laminated Li|PEO|Li pouch cells, and then the area expanded rapidly during cycling. The PPV decreased from 0.11 to 0.06 V. This was caused by the poor interfacial contact and increased polarization (Fig. 4a–c and Supplementary Fig. 20). In stark contrast, the integrated Li@SPE was prepared by coating the electrolyte onto a Li anode and drying in an argon-filled glove box (Supplementary Fig. 21). Upon adhering the two Li@SPEs together, the overpotential of the dual-integrated Li|PTMG-HDI-BHDS|Li pouch cell gradually decreased and then remained stable during the first 10 cycles (Supplementary Fig. 22). The ultrasonic images showed an increased PPV from 0.57 to 0.60 V, along with a gradual change in color from green to yellow, indicating that the interfaces of the integrated Li@SPE were completely self-healed by the dynamic disulfide bonds and hydrogen bonds to form a stable and dense SEI film (Fig. 4d–f).

Based on the above conclusions, Li|SPEs|Li symmetric coin cells were assembled to study the cycling performance (Fig. 4g, 4 h and Supplementary Fig. 23). The Li|PTMG-HDI-BHDS|Li cell assembled with two Li@SPEs exhibited stable voltage hysteresis for more than 6000 h (3000 cycles) at a current density of 0.2 mA cm$^{-2}$, indicating excellent interfacial compatibility and electrochemical stability that exceeded those of many reported SPEs (Supplementary Table 1). The rates of the integrated Li|PTMG-HDI-BHDS|Li cells were tested by cycling from 0.1 to 1.0 mA cm$^{-2}$ with an areal capacity ranging from 0.5 to 5.0 mAh cm$^{-2}$

(Fig. 4h). The lowest overpotentials of 49, 74, 113, 172, 233, and 262 mV were obtained at current densities of 0.1, 0.2, 0.4, 0.6, 0.8, and 1.0 mA cm$^{-2}$, respectively, mainly due to the high ionic conductivity and stable self-healing interface formed in the cell. In sharp contrast, the Li|PEO|Li, Li|PTMG-HDI|Li and laminated Li|PTMG-HDI-BHDS|Li cells showed abrupt failure at a current density of 0.4 mA cm$^{-2}$. The uniform morphology of the Li deposition layer was achieved in the Li|PTMG-HDI-BHDS|Li cells even at a high test rate. In comparison, top-view and cross-sectional SEM images showed irregular dendrite-like domains on the surfaces of the Li anodes with PEO/LiFSI and PTMG-HDI/LiFSI (Supplementary Fig. 24 and Supplementary Fig. 25).

XPS was performed by Ar$^+$ sputtering for 0, 50, 100, and 200 s on the Li-anodes after the rate test to investigate the chemical compositions of the SEI layers. As shown in Fig. 4i and 4j, the F 1s XPS data indicated a high concentration of organic C-F (72%) with a binding energy 688.1 eV and less inorganic Li-F (28%) with a binding energy 685.0 eV was observed in the Li|PEO|Li. With increasing etching depth, the organic content was obviously higher than that of Li|PTMG-HDI-BHDS|Li due to the continuous decomposition of the unstable electrolyte. The nonuniform organic-inorganic compositions also induced highly inhomogeneous stress and strain when the electrode volume expanded during cycling. Consequently, repeated breaking and reforming of the SEI led to low Coulombic efficiency and poor stability.

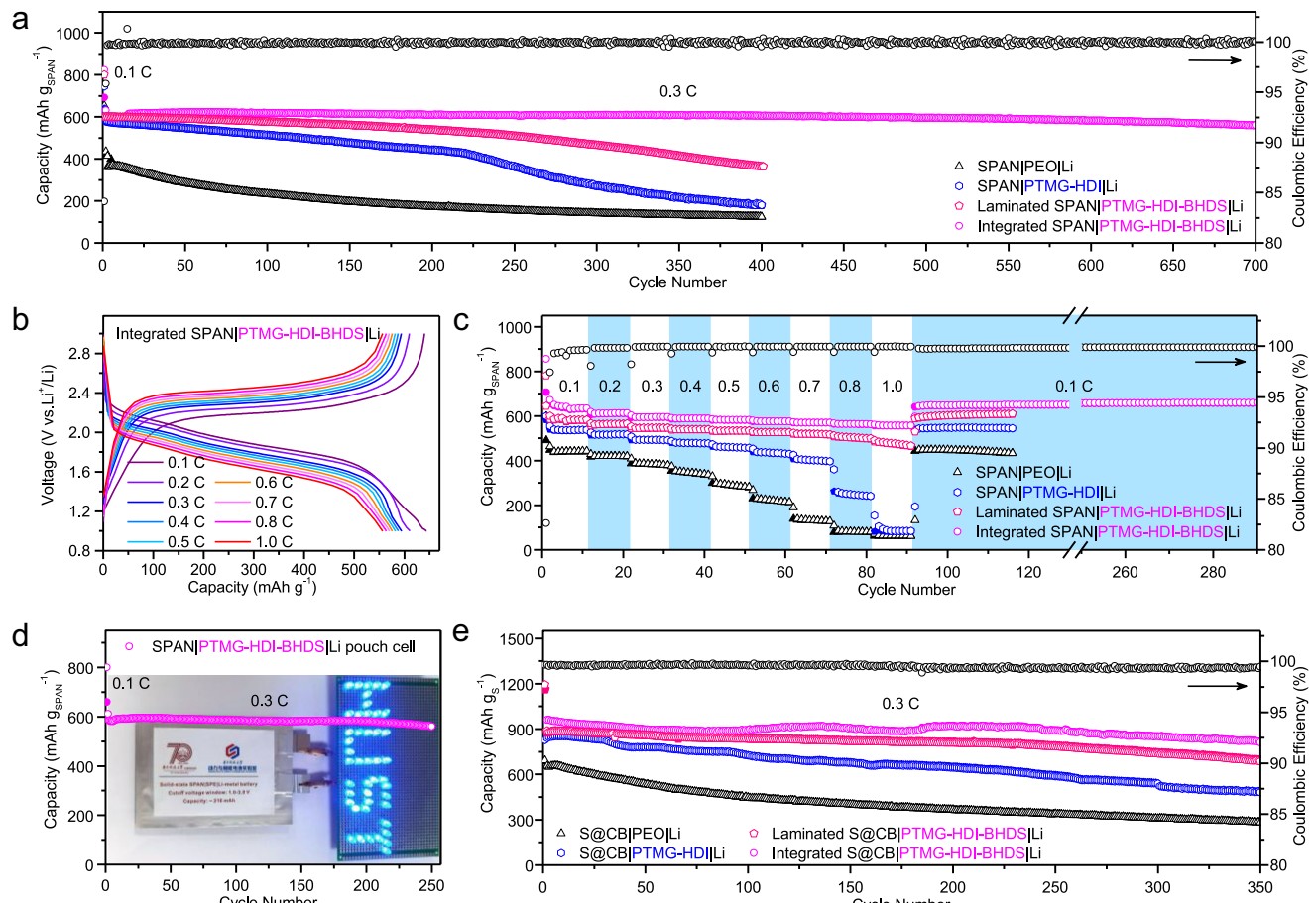

**Fig. 5 | Electrochemical performance of the solid-state Li-S batteries. a** Long-term cycling performance of SPAN|SPEs|Li at 0.3 C. **b** Charge/discharge curves for the SPAN|SPEs|Li cell at various rates. **c** Rate performance of SPAN|SPEs|Li cells. **d** Cycling performance and optical images (inset) of the as-fabricated SPAN|SPEs|Li pouch cell. **e** Cycling performance of S@CB|SPEs|Li at 0.3 C. All tests were performed at 30 °C.

For the Li anode in PTMG-HDI-BHDS/LiFSI, a small C-F content (6%) and a significantly higher Li-F content (94%) were observed before etching, and this became more obvious during deep etching. As previously reported, LiF is an excellent electronic insulator with high interfacial energy for Li⁺ transfer, which enables the migration of Li⁺ through the SEI and stabilizes the interface with the Li anode[29,57–59]. This suggested that the decomposition of the SPE was suppressed and that the decomposition of LiFSI contributed greatly to the formation of SEI.

### Electrochemical Performance of the solid-state Li-S batteries

A PTMG-HDI-BHDS/LiFSI solution was used to wet the anode and cathode (Supplementary Fig. 26). The precursor solutions were cast onto the surface of the cathodes, and integrated SPAN@SPE and Li@SPE were formed simultaneously after solvent evaporation (Supplementary Fig. 27 and Supplementary Fig. 28). The SPAN|PTMG-HDI-BHDS|Li cells were assembled by simply fitting the integrated SPAN@SPE with Li@SPE. As shown in Fig. 5a and Supplementary Fig. 29, after activation at 0.05 C (1 C = 1000 mAh $g_{SPAN}^{-1}$) between 1.0 and 3.0 V, SPAN|PTMG-HDI-BHDS| Li delivered an initial discharge capacity of 602 mAh $g_{SPAN}^{-1}$ at 0.3 C. After 400 cycles, the discharge capacities were 606, 363, 180, and 124 mAh $g_{SPAN}^{-1}$ for the integrated SPAN|PTMG-HDI-BHDS|Li, laminated SPAN|PTMG-HDI-BHDS|Li, integrated SPAN|PTMG-HDI|Li and laminated SPAN|PEO|Li cells, which represented 99.7%, 60.7%, 30.9% and 29.3% of their initial capacities at 0.3 C, respectively. A reversible discharge capacity of 560 mAh $g_{SPAN}^{-1}$ and a Coulombic efficiency above 99% was retained after 700 cycles with the PTMG-HDI-BHDS electrolyte, and the high cycling stability (93% capacity retention) was significantly superior to those of most solid-state Li-S batteries reported thus far

(Supplementary Table 2). It is worth noting that the cells show good consistency and highly reproducible electrochemical performance (Supplementary Fig. 30). The rate performance of the integrated SPAN| SPEs|Li cell was further evaluated (Fig. 5b and 5c). The discharge capacities of the SPAN|PTMG-HDI-BHDS|Li cell from 0.1 to 1.0 C were 641, 611, 594, 588, 582, 576, 570, 563, and 557 mAh $g_{SPAN}^{-1}$, respectively. When the current density was switched back to 0.1 C, a high discharge capacity was recovered and maintained for the next 200 cycles, and the performance was significantly better than those of the laminated SPAN|PTMG-HDI-BHDS|Li, integrated SPAN|PTMG-HDI|Li and laminated SPAN|PEO|Li cells at all tested rates. The SPAN|PTMG-HDI-BHDS|Li cell with a high loading cathode (SPAN: 6.8 mg cm⁻²) delivered a high discharge capacity of 647 mAh g⁻¹ with a high areal capacity of 4.4 mAh cm⁻² at 0.1 C and remained stable over 110 cycles (Supplementary Fig. 31). This provides a promising strategy for the design of high-energy-density solid-state Li-S batteries. The cycling performance of the SPAN|PTMG-HDI-BHDS|Li pouch cell was investigated. A digital photograph shows that the cell battery powered an LED lamp at room temperature (Fig. 5d and Supplementary Fig. 32). The total discharge capacity in the first cycle was as high as 216 mAh (611.8 mAh $g_{SPAN}^{-1}$) at 0.3 C, and the pouch cell exhibited a discharge capacity of 561 mAh $g_{SPAN}^{-1}$ with 91.7% capacity retention after 250 cycles.

To demonstrate the feasibility of using PTMG-HDI-BHDS/LiFSI to avoid polysulfide shuttling and Li dendrite formation, a S-based cathode (S@CB) was prepared with commercial carbon black (CB) as the sulfur host, which exhibits significant capacity attenuation in liquid electrolytes[11]. The capacities and cycling stabilities of the Li-S cells with 2.0 mg cm⁻² of sulfur were evaluated at a current density of 0.3 C

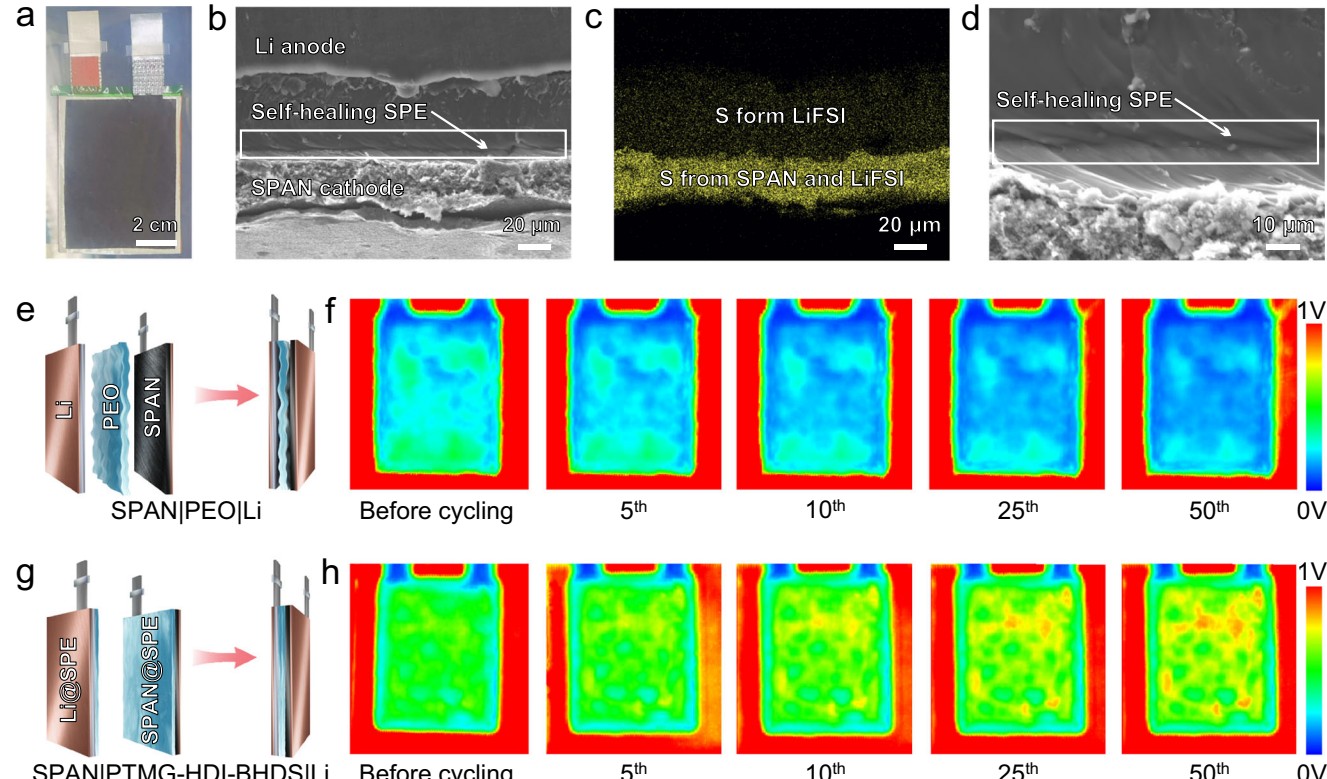

**Fig. 6 | Stable sulfur-cathode|electrolyte|Li-anode interfaces of solid-state Li-S batteries evaluated in pouch cells. a** Optical photographs of the integrated pouch cell. **b, c** Cross-sectional SEM image (**b**) and the corresponding EDS image (**c**) of integrated SPAN|PTMG-HDI-BHDS|Li cells. **d** Magnified SEM image of the self-healing interface. **e** Structure of the SPAN|PEO|Li pouch cell. **f** In situ ultrasonic transmission images of SPAN | PEO|Li during the first 50 cycles. **g** Structure of the SPAN|PTMG-HDI-BHDS|Li pouch cell. **h** In situ ultrasonic transmission images of SPAN|PTMG-HDI-BHDS|Li during the first 50 cycles.

(1 C = 1675 mA $g_S^{-1}$), and the specific capacities were calculated based on the weight of sulfur. All of the cells were first activated at 0.05 C for one cycle (Fig. 5e). The laminated S@CB|PEO|Li, integrated S@CB|PTMG-HDI|Li, and laminated S@CB|PTMG-HDI-BHDS|Li cells exhibited capacities of 282, 483 and 686 mAh $g_S^{-1}$ with 43.0%, 57.4% and 71.3% capacity retention rates after 350 cycles, respectively. The poor cycling stabilities of these cells were caused by ineffective interfacial contact and low ionic conductivity. The shuttling effect remains a serious problem with PEO-based electrolytes because of their molecular structures are similar to those of ether-based electrolytes[60]. The S@CB|PTMG-HDI-BHDS|Li cell showed the highest initial capacity (962 mAh $g_S^{-1}$) after activation, and a reversible capacity of 812 mAh $g_S^{-1}$ (ca. 1.6 mAh cm$^{-2}$) was retained with 84.4% capacity retention after 350 cycles at 0.3 C (Supplementary Fig. 33). PTMG-HDI-BHDS/LiFSI with a relatively low concentration of ether-oxygen suppressed the shuttling of polysulfides, consistent with the high capacity retention (Supplementary Fig. 34).

### Mechanistic analysis for the stable interfaces in solid-state pouch cells

Cross-sectional SEM images and EDS mapping of pouch cells indicated that SPAN@SPE and Li@SPE layers were well integrated without any visible delamination (Fig. 6a–c), which confirmed the utility of dual-integrated electrodes exhibiting interfacial self-healing in solid-state Li-S batteries. PTMG-HDI-BHDS/LiFSI served as both the electrolyte and Li$^+$ conductor in SPAN@SPE, the S was mainly derived from SPAN and LiFSI, and LiFSI was distributed in both the SPAN@SPE layer and the Li@SPE layer, providing abundant channels for the transmission of Li$^+$ ions[4]. The self-healing interface was clearly shown in the enlarged SEM images and backscattered-electron SEM images (Fig. 6d and Supplementary Fig. 35), visually illustrating the 3D interpenetrating Li$^+$ channel between SPAN@SPE and Li@SPE. Systematic EIS studies were

performed, and it was found that the ingenious dual-integrated strategy sharply reduced the interfacial contact resistance and enabled rapid transmission of Li$^+$ (Supplementary Fig. 36).

Figure 6e illustrates the assembly process for the laminated SPAN|PEO|Li pouch cell. Compared with the soft Li anode, the uneven cathode exhibited more severe interface problems, and this contrast was evaluated with ultrasonic imaging. The region of the Al-plastic film appeared red with a PPV of 0.89 V. For the laminated SPAN|PEO|Li, the poor contact at the PEO/LiFSI interface dramatically decreased the ultrasonic signal, resulting in an obvious large dark blue area in ultrasonic image with a PPV of 0.05 V (Fig. 6e and 6f). In subsequent cycles, the blue area gradually expanded to cover the entire cell area due to continuous and uneven deposition of Li. In stark contrast, the integrated SPAN|PTMG-HDI-BHDS|Li showed a uniform green color in the initial state and a gradual increase in PPV from 0.58 to 0.65 V in the next 50 cycles; the resulting color change from green to yellow (Fig. 6g and 6h) indicated a gradually improved contact interface in the SPAN|PTMG-HDI-BHDS|Li cell during cycling.

## Discussion

In summary, we developed a facile route with which to construct poly(ether-urethane)-based electrolytes with self-healing properties, which were cast directly on electrode films to construct integrated solid-state electrodes/electrolytes. Due to rearrangements of the dynamic covalent disulfide bonds and the hydrogen bonds between urethane groups, the interface of two integrated electrodes exhibited self-healing to overcome the problem of multiple interfacial contacts seen with SSLMBs. The constructed SPEs exhibited robust mechanical strength, excellent ion conductivity, a high Li$^+$ transference number, and a broad electrochemical stability window. The long-term cycling stability of a Li|PTMG-HDI-BHDS|Li symmetric cell was investigated for

more than 6000 h. Remarkably, the enhanced interfacial contact provided extremely long cycling stabilities and high capacity retention rates for the SPAN and S@CB cathodes. The dual-integrated strategy by using dynamic disulfide bonds-based self-healing SPEs opens a promising approach for constructing high-performance SSLMBs.

## Methods

### Materials

All reagents were commercially available and used as supplied without further purification. Dimethylacetamide (DMAc), dichloromethane, PEO, polytetrahydrofuran (PTMG, 2000 g mol$^{-1}$), hexamethylene diisocyanate (HDI) and 2-Hydroxyethyl disulfide (BHDS) and dibutyltin dilaurate (DBTDL) catalyst were purchased from Alfa Aesar. Lithium bis(fluorosulfonyl)imide (LiFSI) were purchased from Aldrich.

### Preparation of PTMG-HDI

PTMG-HDI was synthesized via condensation of PTMG with HDI. Briefly, 2.5 g (1.25 mmol) of PTMG and 250 μL (1.5 mmol) of HDI were dissolved in 5 mL of CH$_2$Cl$_2$. 5 μL of DBTDL was added to the solution as a reaction catalyst. The final molar ratio of the monomers (PTMG:HDI) was 1:1.2. The mixture was stirred at 40 °C for 3 h in a N$_2$-filled glove box. Then, 5 mL of DMAc was added to the reaction mixture every 2 h 3 times repeated 3 times, and stirred for 12 h. Finally, a transparent PTMG-HDI solution was obtained.

### Preparation of PTMG-HDI-BHDS

The synthesis of PTMG-HDI-BHDS involved the introduction of the chain extender BHDS into the PTMG-HDI. Briefly, 2.5 g (1.25 mmol) of PTMG, 500 μL (3.0 mmol) of HDI, and 5 μL of DBTDL were dissolved in 5 mL of CH$_2$Cl$_2$ and stirred at 40 °C for 1 h in a N$_2$-filled glove box, and then 200 μL (1.25 mmol) of BHDS was added to the above solution. The final molar ratio of the monomers (PTMG:HDI:BHDS) was 1:2.4:1. Subsequently, 5 mL of DMAc was added into the reaction mixture every 2 h and repeated 3 times, and stirred for 12 h to obtain the PTMG-HDI-BHDS solution.

### Preparation of the PEO/LiFSI, PTMG-HDI/LiFSI, and PTMG-HDI-BHDS/LiFSI electrolytes

Poly(ethylene oxide) (PEO, Mw = 6 × 10$^5$ g mol$^{-1}$) and lithium bis(fluorosulfonyl)imide (LiFSI) were dissolved in acetonitrile with a mass ratio of 2:1 (PEO: LiFSI) to form a uniform PEO/LiFSI solution. The PTMG-HDI and PTMG-HDI-BHDS polymer solutions were mixed with LiFSI in dimethylacetamide (DMAc) at a mass ratio of 2:1 (polymer: LiFSI) and stirred for 3 h. These viscous solutions were degassed under vacuum, cast into a Teflon mold, and then dried in a vacuum oven at 65 °C for 48 h to obtain the PEO/LiFSI, PTMG-HDI/LiFSI, and PTMG-HDI-BHDS/LiFSI electrolytes.

### Preparation of SPAN and S@CB

SPAN was synthesized according to a modified method previously reported[61]. A total of 1.0 g of polyacrylonitrile (PAN) and 3.5 g of S powder were uniformly mixed in 10 mL of ethanol by high-energy ball milling for 4 h. Then, SPAN with a sulfur content of 45.0 wt% was obtained after heating the composite at 300 °C for 450 min in a N$_2$ atmosphere. The S@CB composite with a sulfur content of 80 wt% was prepared by the melt diffusion method[11].

### Material characterization

The SEM analysis was carried out on FEI Nova NanoSEM450 microscope operated at 10 kV. FTIR spectra were measured using a Nicolet iS50 FT/IR Spectrometer (ThermoFisher) with a diamond-attenuated total reflectance attachment. Mechanical testing was carried out on an Instron 5565 testing stain using a strain rate of 5 cm min$^{-1}$. Powder X-ray diffraction (PXRD) analysis was characterized by X-ray powder diffraction (XRD, Cu Ka radiation, Rigaku Ultima IV) with a scanning

rate of 2° min$^{-1}$. XPS tests were performed on ThermoFisher Thermo Scientific KAlpha+ . $^1$H NMR and $^{19}$F NMR spectra of all samples were collected on a Bruker 500 NMR MHz, CDCl$_3$ and DMSO-$d$6 were used as a deuterated solvent and tetramethylsilane as an internal reference, respectively. All samples were dissolved in the solvent to form transparent solutions with a concentration of 10 mg mL$^{-1}$. The XPS testing was carried out using a Physical Electronics Quantera scanning X-ray microprobe, and XPS depth profiling was conducted using argon sputtering. To avoid side reactions or electrode contamination with ambient oxygen and moisture, cycled Li anodes were obtained by disassembling CR2032 coin cells in the glove box with O$_2$ and H$_2$O level <0.1 ppm and then washing them with dimethyl ether (DME). Li metal samples were transported from the glovebox to the SEM and XPS instruments in a hermetically sealed container filled with argon gas, all XPS data were normalized relative to the C $1s$ binding energy of 284.7 eV. Gel permeation chromatography (GPC) was carried out in DMF on two PolyPore columns (Agilent) connected in series with a DAWN multiangle laser light scattering (MALLS) detector.

### Electrochemical characterization

The ionic conductivities of the solid composite electrolytes at different temperatures were measured by sandwiching the SPEs between two steel sheets in CR2032 cells and the frequency range was 0.1–10$^6$ Hz. The ionic conductivity σ was calculated by the equation σ = L/(RS), where R is the total resistance of the solid electrolyte, L is the sample thickness, and S is the area of the solid electrolyte. The Li$^+$ transference number ($t_{Li}^+$) was obtained using a symmetric Li‖Li cell by a chronoamperometry test with a DC voltage amplitude of 50 mV. The EIS tests were taken before and after DC polarization. The $t_{Li}^+$ can be calculated according to $t_{Li}^+ = I_s(\Delta V − R_0I_0)/[I_0(\Delta V − R_sI_s)]$, where $I_0$ and $I_s$ are the initial and steady-state DC current, respectively; $R_0$ and $R_s$ are the initial and steady-state interfacial resistances, respectively. The electrochemical stabilities of the SPEs were evaluated at a scan rate of 10 mV s$^{-1}$ from 0 to 6 V at room temperature.

For electrode fabrication, a cathode slurry was obtained by mixing SPAN or S@CB, CB, SPEs, and PVDF with a weight ratio of 80:10:5:5 in DMAc. The as-obtained slurry was then coated onto a carbon-coated Al foil and dried at 65 °C in a vacuum to give a mass loading of 2.0-2.2 mg cm$^{-2}$ SPAN or sulfur. The high loading of thick SPAN cathodes was 6.8 mg cm$^{-2}$. The integrated cathode@SPEs and Li@SPEs were prepared according to the reported literature by casting the aforementioned polymer electrolyte solutions onto the cathodes and drying under vacuum at 65 °C for 48 h[52]. For the pouch cells, the integrated cathode and anode were punched into rectangles with dimensions of 50 mm × 60 mm, the tab and the Al-plastic film were purchased from Canrd Technology Co. Ltd. A steel mold was also used to provide an external pressure at approximately 200 KPa to the pouch cell. The 2032-type coin cells and pouch cells were assembled in an argon-filled glove box (H$_2$O < 0.1 ppm and O$_2$ < 0.1 ppm). A CHI760E electrochemical workstation was used to test the intrinsic electrochemical properties. Galvanostatic measurements for coin-type and pouch-type cells were conducted with a NEWARE battery test instrument within a cut-off voltage window of 1.0-3.0 V. In the absence of special instructions, all tests were performed at 30 °C in a thermostatic test chamber.

### DFT calculation

Density Functional Theory (DFT) calculations were carried out to better understand the electrolyte. This simulation calculation was carried out on Material Studio software. All computations of electrolytes were calculated by using the DMol$^3$ module. The exchange correlation function was treated by the generalized-gradient-approximation (GGA) with the Perdew-Burke-Ernzerhof (PBE) functional[62]. For the geometry optimization, convergence energy was set to 1.0 × 10$^{-5}$ Ha for energy, 0.001 Ha/Å for force, and 0.005 Å for displacement.

The binding energy (Eb) was computed in the following equation[63]:

$$\Delta E_b = E_{total} - E_{Li^+} - E_{Polymer}$$

where $\Delta E_b$ is the binding energy between the Li$^+$ ion and polymer. $E_{total}$ is the DFT total energy of the corresponding system with adsorbed Li$^+$. $E_{Li^+}$ and $E_s$ represent the Gibbs energies of a Li$^+$ and the corresponding polymer substrates, respectively.

## Data availability

The experiment data that support the findings of this study are available from the corresponding authors upon reasonable request. Source data are provided with this paper.

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

## Acknowledgements

This work is supported by the National Key R&D Program of China (Grant No. 2021YFB2400300 (Z.L.)), the National Natural Science Foundation of China (Grant Nos. 52202236 (F.P.), 5202780089 (Y.H.), 52231009 (Y.H.) and 51972131 (Z.L.)), and China Postdoctoral Science Foundation (Grant No. 2022M711232 (F.P.)). The authors thank the Analytical and Testing Center of Huazhong University of Science and Technology (HUST) and the State Key Laboratory of Materials Processing and Die & Mold Technology for characterizations. Thanks, eceshi (www.eceshi.com) for the XPS test.

## Author contributions

F.P., Z.L., and Y.H. (Yunhui Huang) conceived and designed the research; F.P., L.W., and Y.Z. performed the experiments and the characterization of the materials; Y.L. and Y.H. (Yan Han) conducted part of the characterizations; H.Z., Q.K., and Y.S. performed part of the synthesis and electrochemical tests; H.X. made instructive advice to revise the full text and supported the perfection of the manuscript; F.P., H.X., Z.L., and Y.H. (Yunhui Huang) analyzed all the data and co-wrote the paper. All authors contributed to the discussion of the manuscript.

## Competing interests

The authors declare no competing interests.
