## [Peer Review File · Nature Communications]

Interfacial self-healing polymer electrolytes for long-cycle solid-state lithium-sulfur batteriesREVIEWER COMMENTS

Reviewer #1 (Remarks to the Author):

The authors report self-healing solid polymer electrolytes based on Poly(ether-urethane) by utilizing in-situ polymerization between urethane units and covalent disulfide bonds. By directly casting a modified solid polymer electrolyte (SPE) onto the electrodes, denoted as dual integration, the authors have fabricated a lithium-sulfur cell with improved interfacial contact between the electrode and SPE compared to conventional lamination-based fabrication. The cell exhibited long-term cycling performance of up to 700 cycles with high retention compared to the Li-S cell using conventional lamination of PEO-based SPE. The improved interfacial contact checked by the ultrasound imaging technology, relatively superior mechanical and physical properties, ion conductivity, Li⁺ transference number, and electrical stability was confirmed when compared to PEO-based SPE.

Multiple reports on self-healing solid polymer electrolytes (SPE) for lithium metal anodes exist. Disulfide bonds are also commonly used in self-healing systems (e.g., *Polym. Chem.*, 2022, 13, 6002-6009, *J. Mater. Chem. A.*, 2023, 11, 6503-6521; *Materials Today Energy*, 2022, 24, 100939.). The novelty of this paper lies in using a dually integrated method to develop good interfacial contact between the electrode and SPE. Two coated electrodes are combined using self-healing to fabricate Li-S cells. It is essential to compare the difference between conventional lamination methods and dual integration using the same materials. However, electrochemical performances are compared with PEO-LiFSI, which was prepared from lamination. Only the EIS result was compared with the same materials with different fabrication approaches. Thus, the control group is inadequately selected to show suitability for the dual-integrated and self-healing aspects. As shown in Fig.2f, stretchability differs when the polymers are self-healed, indicating the difference between laminated SPE and SPE formed by the self-healing from two coated electrodes. In addition, crucial information such as lithium salt concentration optimization, mechanical comparison between original and self-healed samples, and polysulfide shuttle effect prevention (e.g., H-Cell test) is not fully provided to support this work. Thus, the manuscript can be evaluated for publication suitability when the issues related to scientific advances are convinced.

1. The improvement in electrochemical performance is primarily attributed to the material selection rather than the fabrication approach. The electrochemical performance of the cells using PTMG-HDI-BHDS should be compared with the different fabrications, which are dual-integrated and laminated (Fig. 4h, 5a, 5c, and 5e).
2. EIS plots and circuit models with different SPEs and fabrication are required.
3. As indicated in the Introduction, In-situ polymerization is uncontrollable. It is necessary to compare molecular weight-viscosity data for PTMG-HDI in addition to PTMG-HDI-BHDS in Fig. 2, which shows pot life. What is the storage life of the PTMG-HDI-BHDS? Is it stable for more than 350 minutes?
4. Stress-strain measurements and FT-IR analysis of original and self-healed PTMG-HDI-BHDS are required to compare the effect of self-healing on the performances.
5. Optimization of the lithium salts or the effect of salt concentration is necessary. Do the PEO-LiFSI and PTMG-HDI-LiFSI show optimization with the same concentration of self-healing SPE?

Reviewer #2 (Remarks to the Author):

The study titled “Interfacial self-healing polymer electrolytes for ultralong-life solid-state lithium-sulfur batteries” explores the utilization of self-healing polymer electrolytes in all-solid-state batteries, showing remarkable electrochemical performance. The authors have conducted a thorough investigation and provides valuable contributions to the field. The reviewer presents the following concerns. Kindly provide an appropriate response to address these issues.

- Other studies of the self-healing concept with an aliphatic chain structure similar to this work have been published by applying other cathode materials, such as LFP. (Nat. Commun., 2022, 13, 4868 / Macromolecules, 2020, 53, 1024 / Polymer Chemistry, 2022, 13, 6002 et al) Can the authors claim any novelty in their polymer electrolyte research, aside from the different cathode active materials?
- Insufficient compositional information is provided regarding the ratio of components used in the preparation of polymer electrolytes and the final product. This information is essential in terms of offering valuable insights to readers as well.
- The loading level of the sulfur electrodes in this study is rather low (2~2.2 mg cm⁻²). Based on the discharge capacity of the sulfur electrode (600 mAh g⁻¹), the current density is only about 1.3 mAh cm⁻². This is a significantly lower value than LIB, which uses a current density of about 4 to 5 mAh cm⁻².
- To achieve the long cycle life described by the authors, it is necessary to apply sufficient external pressure to the pouch cell containing the Li metal anode. Please provide an explanation for this requirement and ensure it is adequately reflected in the manuscript.
- In this manuscript, authors used too many ‘interface impedance’ or ‘interfacial impedance’. But, in my opinion, interface impedance is not an appropriate expression in electrochemistry. I recommend using the word ‘resistance’ instead of ‘impedance’.
- The authors suggest that the polymer electrolyte developed by them has a wide potential window and is particularly stable at high potentials, but such a wide potential window is meaningless when using a sulfur cathode. (In this study, the potential is only used up to 3V vs Li/Li+). If the authors want to appeal to a wide potential window, add experiments on cathode materials that react at high potentials, otherwise move the related content to the supporting information.
- Line 87: What is SPAN? Provide its full name.
- Line 103: What is polymerized? Is it polymerized? Please check it.
- Apart from the aforementioned errors, numerous typographical errors have been identified. The manuscript contains mistakes and inconsistencies in subscript notation and the usage of symbols. Therefore, it is necessary to edit the manuscript and submit relevant corrections and certificates.
- Fig. 3: The axis titles in Fig. 3a are wrong. Please correct it.

Reviewer #3 (Remarks to the Author):

Comments on article number: NCOMMS-23-20631

Title: Interfacial self-healing polymer electrolytes for ultralong-life solid-state lithium-sulfur batteries

Article Type: Communication

Corresponding Authors: Z. Li, Y. Huang

Authors: F. Pei, L. Wu, Y. Zhang, Y. Liao, Q. Kang, Y. Han, H. Zhang, Y. Shen, H. Xu, Z. Li, Y. Huang

The paper proposes a cleverly engineered solid polymer electrolyte based on a poly ether-urethane capable of self-healing. Although the approach proposed is not new in concept (see for example DOI:10.3390/polym15051145 or DOI:10.1007/s40820-023-01075-9), it clearly shows promise in terms of processability and scalability for commercial Li-S applications.

This reviewer has identified a few points worth a more in-depth discussion mainly on some chemical related aspects and on cell processing.

Concerning the PTMG oligomer, the authors present results for the average $M_w=2000$. However, from a cursory look at the chemical providers, it will be very easy to obtain various lengths ranging from very short $M_w=250$ up to longer $M_w=3000$ and beyond. What is the rationale behind the specific polymer length? Can the authors provide data for very short and much longer oligomers? What are the critical effects PTMG length has on the solid electrolyte characteristics?

The polymer electrolyte (including LiFSI) has a relatively low conductivity at RT but an impressively high transference number (0.81). This is a very helpful trait for high-rate capabilities as demonstrated in Fig 5. However, there is nothing immediately apparent justifying such a high transference number as there are no elements that would trap/hinder the FSI- anion, Can the authors explain the origin and mechanism by which such a high value is obtained?

The other important aspect of this manuscript is the cell handling, and the fabrication of meaningful 5x6 cm footprint cells is very laudable and an encouraging sign. Many more points appear on this front. The polymer solution appears to be very viscous so one interesting aspect is cathode permeation. Is there a critical thickness (say no more than 20 μ m) beyond which cathode permeation is failing (the polymer does not reach the bottom layer close to the current collector)? Do the authors have data on thicker cathodes (even if energy density is not a problem as the aim of the manuscript is the S cathode)? Similarly, for full permeation of the standard cathode, is there a critical porosity value? Do the authors have data on porosity as it is not currently treated in version of the work? The situation can be roughly inferred by Fig 6 and Supp Fig 20 but can the authors provide backscattered electrons imaging as permeation is very hard to judge with the current pictures.

Continuing on the polymer solutions, the mixing described in the methods section will yield roughly 3g of electrolyte. Is it possible to mix bigger quantities? What is the stable shelf life of the mixture and are there any special conditions (say fridge storage for example)? How many 5x6cm cells can be coated with a batch of 3g? Did the authors investigate mixing larger batches (say 50g, 100g)?

Lastly, a few points concerning the cells themselves: for obvious reasons, the authors are presenting a very limited set of data concerning a few cells but is there statistical significance in the presented data? How many cells were prepared, and what is the reproducibility ratio? Can the authors provide a figure on statistical significance?

In comparing Supp Fig 20 and 21, it looks like the electrolyte layer is $\sim 10\mu$ m thick while in Fig 6c it is more like 20-30 μ m. By what method was the 10 μ m deposited? Is there a critical thickness below which short circuit cannot be avoided and after which energy density is affected at the cell level? What is the authors recommendation on the subject if these cells were to be put into a production line?

And speaking of production lines, when using laminated Cu/Li foils for the anode, it is critical to quality check the surface of the foil for any imperfections and impurities. Did the authors take any measure to control the initial Li surface features? What's their experience of the effect of such features on their cells?

Reviewer Comments

Reviewer #1 (Remarks to the Author):

The authors report self-healing solid polymer electrolytes based on Poly(ether-urethane) by utilizing in-situ polymerization between urethane units and covalent disulfide bonds. By directly casting a modified solid polymer electrolyte (SPE) onto the electrodes, denoted as dual integration, the authors have fabricated a lithium-sulfur cell with improved interfacial contact between the electrode and SPE compared to conventional lamination-based fabrication. The cell exhibited long-term cycling performance of up to 700 cycles with high retention compared to the Li-S cell using conventional lamination of PEO-based SPE. The improved interfacial contact checked by the ultrasound imaging technology, relatively superior mechanical and physical properties, ion conductivity, Li⁺ transference number, and electrical stability was confirmed when compared to PEO-based SPE.

Multiple reports on self-healing solid polymer electrolytes (SPE) for lithium metal anodes exist. Disulfide bonds are also commonly used in self-healing systems (e.g., *Polym. Chem.*, 2022, 13, 6002-6009, *J. Mater. Chem. A.*, 2023, 11, 6503-6521; *Materials Today Energy*, 2022, 24, 100939.). The novelty of this paper lies in using a dually integrated method to develop good interfacial contact between the electrode and SPE. Two coated electrodes are combined using self-healing to fabricate Li-S cells. It is essential to compare the difference between conventional lamination methods and dual integration using the same materials. However, electrochemical performances are compared with PEO-LiFSI, which was prepared from lamination. Only the EIS result was compared with the same materials with different fabrication approaches. Thus, the control group is inadequately selected to show suitability for the dual-integrated and self-healing aspects. As shown in Fig.2f, stretchability differs when the polymers are self-healed, indicating the difference between laminated SPE and SPE formed by the self-healing from two coated electrodes. In addition, crucial information such as lithium salt concentration optimization, mechanical comparison between original and self-healed samples, and polysulfide shuttle effect prevention (e.g., H-Cell test) is not fully provided to support this work. Thus, the manuscript can be evaluated for publication suitability when the issues related to scientific advances are convinced.

1. The improvement in electrochemical performance is primarily attributed to the material selection rather than the fabrication approach. The electrochemical performance of the cells using PTMG-HDI-BHDS should be compared with the different fabrications, which are dual-integrated and laminated (Fig. 4h, 5a, 5c, and 5e).

2. EIS plots and circuit models with different SPEs and fabrication are required.

3. As indicated in the Introduction, In-situ polymerization is uncontrollable. It is necessary to compare molecular weight-viscosity data for PTMG-HDI in addition to PTMG-HDI-BHDS in Fig. 2, which shows pot life. What is the storage life of the PTMG-HDI-BHDS? Is it stable for more than 350 minutes?

4. Stress-strain measurements and FT-IR analysis of original and self-healed PTMG-HDI-

BHDS are required to compare the effect of self-healing on the performances.

5. Optimization of the lithium salts or the effect of salt concentration is necessary. Do the PEO-LiFSI and PTMG-HDI-LiFSI show optimization with the same concentration of self-healing SPE?

Reviewer #2 (Remarks to the Author):

The study titled “Interfacial self-healing polymer electrolytes for ultralong-life solid-state lithium-sulfur batteries” explores the utilization of self-healing polymer electrolytes in all-solid-state batteries, showing remarkable electrochemical performance. The authors have conducted a thorough investigation and provides valuable contributions to the field. The reviewer presents the following concerns. Kindly provide an appropriate response to address these issues.

1- Other studies of the self-healing concept with an aliphatic chain structure similar to this work have been published by applying other cathode materials, such as LFP. (Nat. Commun., 2022, 13, 4868 / Macromolecules, 2020, 53, 1024 / Polymer Chemistry, 2022, 13, 6002 et al) Can the authors claim any novelty in their polymer electrolyte research, aside from the different cathode active materials?

2- Insufficient compositional information is provided regarding the ratio of components used in the preparation of polymer electrolytes and the final product. This information is essential in terms of offering valuable insights to readers as well.

3- The loading level of the sulfur electrodes in this study is rather low (2~2.2 mg cm⁻²). Based on the discharge capacity of the sulfur electrode (600 mAh g⁻¹), the current density is only about 1.3 mAh cm⁻². This is a significantly lower value than LIB, which uses a current density of about 4 to 5 mAh cm⁻².

4- To achieve the long cycle life described by the authors, it is necessary to apply sufficient external pressure to the pouch cell containing the Li metal anode. Please provide an explanation for this requirement and ensure it is adequately reflected in the manuscript.

5- In this manuscript, authors used too many ‘interface impedance’ or ‘interfacial impedance’. But, in my opinion, interface impedance is not an appropriate expression in electrochemistry. I recommend using the word ‘resistance’ instead of ‘impedance’.

6- The authors suggest that the polymer electrolyte developed by them has a wide potential window and is particularly stable at high potentials, but such a wide potential window is meaningless when using a sulfur cathode. (In this study, the potential is only used up to 3V vs Li/Li⁺). If the authors want to appeal to a wide potential window, add experiments on cathode materials that react at high potentials, otherwise move the related content to the supporting information.

7- Line 87: What is SPAN? Provide its full name.

8- Line 103: What is polymerized? Is it polymerized? Please check it.

9- Apart from the aforementioned errors, numerous typographical errors have been identified. The manuscript contains mistakes and inconsistencies in subscript notation and the usage of symbols. Therefore, it is necessary to edit the manuscript and submit relevant corrections and certificates.

10- Fig. 3: The axis titles in Fig. 3a are wrong. Please correct it.

Reviewer #3 (Remarks to the Author):

Comments on article number: NCOMMS-23-20631

Title: Interfacial self-healing polymer electrolytes for ultralong-life solid-state lithium-sulfur batteries

Article Type: Communication

Corresponding Authors: Z. Li, Y. Huang

Authors: F. Pei, L. Wu, Y. Zhang, Y. Liao, Q. Kang, Y. Han, H. Zhang, Y. Shen, H. Xu, Z. Li, Y. Huang

The paper proposes a cleverly engineered solid polymer electrolyte based on a poly ether-urethane capable of self-healing. Although the approach proposed is not new in concept (see for example DOI:10.3390/polym15051145 or DOI:10.1007/s40820-023-01075-9), it clearly shows promise in terms of processability and scalability for commercial Li-S applications.

This reviewer has identified a few points worth a more in-depth discussion mainly on some chemical related aspects and on cell processing.

1. Concerning the PTMG oligomer, the authors present results for the average $M_w=2000$. However, from a cursory look at the chemical providers, it will be very easy to obtain various lengths ranging from very short $M_w=250$ up to longer $M_w=3000$ and beyond. What is the rationale behind the specific polymer length? Can the authors provide data for very short and much longer oligomers? What are the critical effects PTMG length has on the solid electrolyte characteristics?

2. The polymer electrolyte (including LiFSI) has a relatively low conductivity at RT but an impressively high transference number (0.81). This is a very helpful trait for high-rate capabilities as demonstrated in Fig 5. However, there is nothing immediately apparent justifying such a high transference number as there are no elements that would trap/hinder the FSI⁻ anion, Can the authors explain the origin and mechanism by which such a high value is obtained?

3. The other important aspect of this manuscript is the cell handling, and the fabrication of meaningful 5x6 cm footprint cells is very laudable and an encouraging sign. Many more points appear on this front. The polymer solution appears to be very viscous so one interesting aspect is cathode permeation. Is there a critical thickness (say no more than 20um) beyond which cathode permeation is failing (the polymer does not reach the bottom layer close to the current collector)? Do the authors have data on thicker cathodes (even if energy density is not a problem as the aim of the manuscript is the S cathode)? Similarly, for full permeation of the standard cathode, is there a critical porosity value? Do the authors have data on porosity as it is not currently treated in version of the work? The situation can be roughly inferred by Fig 6 and Supp Fig 20 but can the authors provide backscattered electrons imaging as permeation is very hard to judge with the current pictures.

4. Continuing on the polymer solutions, the mixing described in the methods section will yield roughly 3g of electrolyte. Is it possible to mix bigger quantities? What is the stable shelf life of the mixture and are there any special conditions (say fridge storage for example)? How many 5x6cm cells can be coated with a batch of 3g? Did the authors investigate mixing larger batches (say 50g, 100g)?

5. Lastly, a few points concerning the cells themselves: for obvious reasons, the authors are presenting a very limited set of data concerning a few cells but is there statistical significance in the presented data? How many cells were prepared, and what is the reproducibility ratio? Can the authors provide a figure on statistical significance?

6. In comparing Supp Fig 20 and 21, it looks like the electrolyte layer is ~10um thick while in Fig 6c it is more like 20-30um. By what method was the 10um deposited? Is there a critical thickness below which short circuit cannot be avoided and after which energy density is affected at the cell level? What is the authors recommendation on the subject if these cells were to be put into a production line?

7. And speaking of production lines, when using laminated Cu/Li foils for the anode, it is critical to quality check the surface of the foil for any imperfections and impurities. Did the authors take any measure to control the initial Li surface features? What's their experience of the effect of such features on their cells?

Point-by-point response (NCOMMS-23-20631)

Reviewer #1

Reviewer #1 (Remarks to the Author): The authors report self-healing solid polymer electrolytes based on Poly(ether-urethane) by utilizing in-situ polymerization between urethane units and covalent disulfide bonds. By directly casting a modified solid polymer electrolyte (SPE) onto the electrodes, denoted as dual integration, the authors have fabricated a lithium-sulfur cell with improved interfacial contact between the electrode and SPE compared to conventional lamination-based fabrication. The cell exhibited long-term cycling performance of up to 700 cycles with high retention compared to the Li-S cell using conventional lamination of PEO-based SPE. The improved interfacial contact checked by the ultrasound imaging technology, relatively superior mechanical and physical properties, ion conductivity, Li⁺ transference number, and electrical stability was confirmed when compared to PEO-based SPE.

Multiple reports on self-healing solid polymer electrolytes (SPE) for lithium metal anodes exist. Disulfide bonds are also commonly used in self-healing systems (e.g., Polym. Chem., 2022,13, 6002-6009, J. Mater. Chem. A., 2023, 11, 6503-6521; Materials Today Energy, 2022, 24, 100939.). The novelty of this paper lies in using a dually integrated method to develop good interfacial contact between the electrode and SPE. Two coated electrodes are combined using self-healing to fabricate Li-S cells. It is essential to compare the difference between conventional lamination methods and dual integration using the same materials. However, electrochemical performances are compared with PEO-LiFSI, which was prepared from lamination. Only the EIS result was compared with the same materials with different fabrication approaches. Thus, the control group is inadequately selected to show suitability for the dual-integrated and self-healing aspects. As shown in Fig.2f, stretchability differs when the polymers are self-healed, indicating the difference between laminated SPE and SPE

formed by the self-healing from two coated electrodes. In addition, crucial information such as lithium salt concentration optimization, mechanical comparison between original and self-healed samples, and polysulfide shuttle effect prevention (e.g., H-Cell test) is not fully provided to support this work. Thus, the manuscript can be evaluated for publication suitability when the issues related to scientific advances are convinced.

Response: We appreciate the reviewer's valuable comments. We have carefully revised our manuscript according to the suggestions, and below is a belief summary of the innovations of our work:

Innovation #1: In the three literatures mentioned by the reviewer, the designed SPEs with disulfide bonds were mainly focused on inhibiting the crack or deformation in the electrode materials during the cycling. However, the large interfacial ionic transfer barriers between the SPEs (without any solvents) and porous electrodes were despised. For example, ionic liquid (IL) and ethylene carbonate (EC) with a high content of 60 wt% was introduced to SPE to reduce the interfacial resistance and promote ionic conductivity (*J. Mater. Chem. A*, 2023, 11, 6503-6521). In our work, two integrated electrodes/SPEs were prepared by the pre-impregnation coating method. The SPE acted as both a Li⁺ conductor and binder, providing 3D interpenetrating Li⁺ pathways in high-loading SPAN cathode (6.8 mg cm⁻² with a thickness of ~80 μm) (see lines 371-376, Supplementary Fig. 31 and 35). Then the disulfide bonds with self-healing character mainly focuses on solving the multiple interfacial contact problem between the SPEs on the surface of the two integrated electrodes@SPEs. The SPAN|PTMG-HDI-BHDS|Li cell delivered a high discharge capacity of 647 mAh g⁻¹ with a high areal capacity of 4.4 mAh cm⁻² and kept stable over 110 cycles (Supplementary Fig. 31). This work provided a promising strategy for the design of high-energy-density solid-state Li-S batteries.

Figure | The role of disulfide bonds in the construction of an integrated battery.

Innovation #2: Although disulfide-bond-based polymers have excellent self-healing ability and viscoelasticity, but suffer from poor mechanical strength. In our work, the terminal hydroxyls of PTMG reacted readily with hexamethylene diisocyanate (HDI), the molecular weight of PTMG-HDI-BHDS were further increased by the skillfully introducing 2-hydroxyethyl disulfide (BHDS) as chain extender to enhance the mechanical strength (Fig 2b, 4g and Supplementary Fig. 3). The breaking strength of PTMG-HDI-BHDS were 88.3 MPa and 2000%, significantly higher than those of PUSS₁PEG₃ (0.47 MPa) (*Polym. Chem.*, 2022), PAES-g-PUS/2PEG90 (2.34 MPa) (*J. Mater. Chem. A.*, 2023) and RFSPE-3 (20 MPa) (*Materials Today Energy*, 2022). As a result, the Li|SPE|Li symmetric cells exhibited stable long-term cycling for 6000 h, indicating excellent ability to suppress Li dendrites growth.

Innovation #3: Because the assembled battery is a black box, there are no effective strategies to track the solid-solid interfacial contact problems in real time. Ultrasound imaging is a powerful tool to evaluate interfacial contacts and side reactions due to its high sensitivity to gas/vacuum. Without disassemble the battery, in situ ultrasonic scanning imaging technique was used to nondestructively verify the key role of disulfide bonds in repairing the electrolyte interfaces (see Fig. 4a-4e and Fig. 6e-6h, ref. 55 and 56).

As suggested by the reviewer, a series of additional experiments and discussions, such as the LiFSI concentration optimization, mechanical comparison and polysulfide shuttling effect, have been provided in detail in the revised manuscript. In particular, observing the diffusion of polysulfides in the liquid or gel electrolyte in H-type cell provides the most intuitive evidence for the study of shuttle effect in traditional Li-S batteries. However, polysulfide solution leads to severe swelling and destruction of the PTMG-HDI-BHDS/LiFSI membrane in H-type cell; this test cannot truly reflect whether shuttle effects occur in this solid-state electrolyte. To address the reviewer's concern, the recycled S@CB|SPEs|Li cells were disassembled (Supplementary Fig. 34), which showed that large amount of dissolved yellow polysulfides adhered on the surface of PEO/LiFSI after cycling. In comparison, no noticeable yellow color was observed on the surface of PTMG-HDI-BHDS/LiFSI. Significantly lower concentration of ether-oxygen structure in PTMG-HDI-BHDS effectively suppressed the shuttling of polysulfides, matching well with the high-capacity retention (see lines 399-402 of page 16, Supplementary Fig. 34).

Supplementary Fig. 34 Digital photographs of the diffusion of polysulfides electrolytes after disassembled the cycled (a) S@CB|PEO|Li and (b) S@CB|PTMG-HDI-BHDS|Li cells.

1. **Comment:** The improvement in electrochemical performance is primarily attributed to the material selection rather than the fabrication approach. The

electrochemical performance of the cells using PTMG-HDI-BHDS should be compared with the different fabrications, which are dual-integrated and laminated (Fig. 4h, 5a, 5c, and 5e).

Response: Thanks for the reviewer's suggestions, the electrochemical performances of PTMG-HDI-BHDS/LiFSI by using dual-integrated and laminated strategies were provided in the revised manuscript (Fig. 4h, 5a, 5c, and 5e). Compared with PTMG-HDI/LiFSI, the designed PTMG-HDI-BHDS/LiFSI in the laminated batteries showed obvious performance advantages. On this basis, the dual-integrated fabrication strategy further improved the cycling stability compared to the laminated method. The additional experiment and the corresponding discussion relevant information have been provided in the revised manuscript (see line 311 of page 13, lines 353-357 of page 15, lines 369-371 of page 15, lines 389-392 of page 16, Fig. 4h, 5a, 5c, and 5e).

2. Comment: EIS plots and circuit models with different SPEs and fabrication are required.

Response: Thanks for the reviewer's suggestions. Systematic EIS studies with circuit models were performed on PEO/LiFSI, PTMG-HDI/LiFSI and PTMG-HDI-BHDS/LiFSI with different fabrication strategies (see lines 283-285 of page 12, lines 425-428 of page 18 and Supplementary Fig. 36). The impedance of these SPAN|SPEs|Li decreased overall in the order of PEO/LiFSI > PTMG-HDI/LiFSI > PTMG-HDI-BHDS/LiFSI, agreeing well with their ionic conductivities. It is worth noting that the dual-integrated strategy cannot be adopted with PEO/LiFSI because the significant side reaction between the solvent of acetonitrile and Li (PEO was dissolved in acetonitrile). For PTMG-HDI/LiFSI and PTMG-HDI-BHDS/LiFSI, as shown in the Supplementary Fig. 36, the dual-integrated strategy remarkably reduced the interfacial resistance between SPEs and electrodes compared to the laminated strategy. Overall, the ionic conductivity of the SPEs plays a crucial and decisive role in improving the

electrochemical performance of the battery.

Supplementary Fig. 36 EIS plots of the (a) SPAN|PTMG-HDI-BHDS|Li, (b) SPAN|PTMG-HDI|Li and (c) SPAN|PEO|Li with different fabrication strategies.

3. **Comment:** As indicated in the Introduction, In-situ polymerization is uncontrollable. It is necessary to compare molecular weight-viscosity data for PTMG-HDI in addition to PTMG-HDI-BHDS in Fig. 2, which shows pot life. What is the storage life of the PTMG-HDI-BHDS? Is it stable for more than 350 minutes?

Response: Thanks for the reviewer's suggestions, the molecular weight-viscosity data of PTMG-HDI was provided in the Supplementary Fig. 3, the viscosity increased synchronously with the molecular weight for 8 h at 40 °C, the molecular weight (M_n) and viscosity (η) of PTMG-HDI were increased to $5.97 \times 10^5 \text{ g mol}^{-1}$ and $0.88 \times 10^4 \text{ cps}$, respectively, significantly lower than that of PTMG-HDI-BHDS (Fig. 2b). The horizontal coordinate in Fig. 2b is the polymerization time, not the storage time. As the reaction proceeded, the molecular weight and viscosity of the polymer reached a constant value and did

not change with the reaction time. We then measured the viscosity and molecular weight of the polymer solution after one month of storage, and these values remained constant, due to the excellent thermal and chemical stability of polyurethane (see lines 149-152 of page 7 and Supplementary Fig. 3).

Supplementary Fig. 3 Variation processes of molecular weight (red point) and viscosity (blue point) during polymerization.

4. **Comment:** Stress-strain measurements and FT-IR analysis of original and self-healed PTMG-HDI-BHDS are required to compare the effect of self-healing on the performances.

Response: Thanks for the reviewer's suggestions, the mechanical properties of the self-healed PTMG-HDI-BHDS film were further investigated to evaluate the self-healing ability (Supplementary Fig. 10a). The break strength and ultimate elongation of the self-healed PTMG-HDI-BHDS were 84.3 MPa and 1920%, respectively, which were very close to the original value, indicating that the self-healed fracture surface was well integrated in the presence of abundant hydrogen bonds and disulfide bonds, and could be restored to the original mechanical strength. The resulting characteristic functional groups of the pristine and self-healed SPEs were compared by ATR-FTIR (Fig. 2g), the infrared characteristic peaks of all functional groups (e.g., -S-S-, -NH-COO-) were well coincident (see lines 185-188 of page 8 and Supplementary Fig. 10b).

Supplementary Fig. 10 (a) Stress-strain measurements and (b) FT-IR analysis of the original and self-healed PTMG-HDI-BHDS.

5. **Comment:** Optimization of the lithium salts or the effect of salt concentration is necessary. Do the PEO-LiFSI and PTMG-HDI-LiFSI show optimization with the same concentration of self-healing SPE?

Response: According to the reviewers' suggestion, the effect of lithium salt (LiFSI) content on ionic conductivity was provided in the revised manuscript. The ratios between the polymer and LiFSI were set to 3:1, 2:1, 1.5:1 and 1:1, we found that under the condition of proper concentration of LiFSI (mass ratio of 2:1), the PTMG-HDI/LiFSI and PTMG-HDI-BHDS/LiFSI were overall in the highest range of ionic conductivity. For the PEO/LiFSI, when the ratio was 1.5:1, the conductivity reached the highest value ($1.23 \times 10^{-5} \text{ S cm}^{-1}$), but was not much different from the conductivity of the ratio of 2:1 ($1.20 \times 10^{-5} \text{ S cm}^{-1}$). High content of LiFSI could not significantly improve the conductivity, but seriously reduced the mechanical strength of the SPEs (see Lines 222-223 of page 10 and Supplementary Fig. 16).

Supplementary Fig. 16 The effect of LiFSI content on ionic conductivity of (a) PEO/LiFSI, (b) PTMG-HDI/LiFSI and (c) PTMG-HDI/LiFSI.

Reviewer 2

Reviewer #1 (Remarks to the Author): The study titled “Interfacial self-healing polymer electrolytes for ultralong-life solid-state lithium-sulfur batteries” explores the utilization of self-healing polymer electrolytes in all-solid-state batteries, showing remarkable electrochemical performance. The authors have conducted a thorough investigation and provides valuable contributions to the field. The reviewer presents the following concerns. Kindly provide an appropriate response to address these issues.

Response: We would like to thank the reviewer’s positive comments. We have carefully revised our manuscript according to the reviewer’s suggestions.

Please also see the detailed replies to the following comments:

1. **Comment:** Other studies of the self-healing concept with an aliphatic chain structure similar to this work have been published by applying other cathode materials, such as LFP. (Nat. Commun., 2022, 13, 4868 / Macromolecules, 2020, 53, 1024 / Polymer Chemistry, 2022, 13, 6002 et al) Can the authors claim any novelty in their polymer electrolyte research, aside from the different cathode active materials?

Response: We appreciate for the reviewer's comments on our paper. We have carefully revised our manuscript according to the suggestions, and below is a belief summary of the innovations in our work, which was also presented in the response to reviewer 1:

Innovation #1: In our work, a new class of poly(ether-urethane)-based SPEs with high mechanical strength and ionic conductivity were propose. From the point of view of molecular structure design, 2-hydroxyethyl disulfide (BHDS) was introduced into PTMG-HDI-BHDS as a chain extender to increase the polymerization degree, endowing the SPE with excellent self-healing ability. and the ether-oxygen and carbonyl functional groups in the structure enable dissociation of the lithium salts and improve the ionic conductivity ($2.4 \times 10^{-4} \text{ S cm}^{-1}$ at 30 °C). Compared with PEO/LiFSI (-CH₂-CH₂-O-), the proposed PTMG-HDI-BHDS/LiFSI (-CH₂-CH₂-CH₂-CH₂-O-) with a relatively low concentration of ether-oxygen suppressed the dissolution of polysulfide into the SPE, matching well with the high-capacity retention (see lines 399-402 of page 16, Supplementary Fig. 34).

Innovation #2: In the three literatures mentioned by the reviewer, the designed SPEs by using disulfide bonds were mainly focused on inhibiting the crack or deformation in the electrode materials during the cycling. However, the large interfacial ionic transfer barriers between the SPEs (without any solvents) and porous electrodes were despised. For example, all test were conducted at high temperature of 60 °C to reduce the interfacial resistance and promote the electrochemical performance (*Macromolecules*, 2020, 53, 1024). In our work,

dual integrated electrodes/SPEs were prepared by the pre-impregnation coating method. Then the disulfide bonds with self-healing character mainly focuses on solving the multiple interfacial contact problem between the SPEs on the surface of the two integrated electrodes@SPEs. In situ ultrasonic scanning imaging technique was further used to nondestructively verify the key role of disulfide bonds in repairing the electrolyte interfaces (see Fig. 4a-4e and Fig. 6e-6h, ref. 55 and 56). The SPE simultaneously acted as both a Li⁺ conductor and binder, providing Li⁺ 3D interpenetrating pathways even in high loading thick SPAN cathode. The SPAN|PTMG-HDI-BHDS|Li cell delivered a high discharge capacity of 647 mAh g⁻¹ with a high areal capacity of 4.4 mAh cm⁻² (6.8 mg cm⁻²) and kept stable over 110 cycles. This work provided a promising strategy for the design of high-energy-density solid-state Li-S batteries. (see lines 371-376 of page 16, Supplementary Fig. 31 and 35).

Innovation #3: Although disulfide-bond-based polymers have excellent self-healing ability and viscoelasticity, but often face poor mechanical strength. In our work, the terminal hydroxyls of PTMG reacted readily with hexamethylene diisocyanate (HDI), the molecular weight of PTMG-HDI-BHDS were further increased by the skillfully introducing 2-hydroxyethyl disulfide (BHDS) as chain extender to enhance the mechanical strength (Fig 2b, 4g and Supplementary Fig. 3). The breaking strength of PTMG-HDI-BHDS were 88.3 MPa and 2000%, significantly higher than those of PUSS₁PEG₃ (0.47 MPa) (*Polym. Chem.*, 2022), DSE-3 (42.6 MPa) (*Nat. Commun.*, 2022) and 3PEG-SSH (0.092 MPa) (*Macromolecules*, 2020). As a result, the Li|SPE|Li symmetric cells exhibited stable long-term cycling for 6000 h, indicating excellent ability to suppress Li dendrites growth.

2. Comment: Insufficient compositional information is provided regarding the ratio of components used in the preparation of polymer electrolytes and the final product. This information is essential in terms of offering valuable insights to

readers as well.

Response: Thanks for the reviewer's suggestions. We have provided more experimental details to describe the component proportions of the final product in the experimental section in the revised manuscript (see lines 465-468 of page 19, lines 474-476 of page 19, lines 481-484 of page 20).

3. Comment: The loading level of the sulfur electrodes in this study is rather low ($2\sim 2.2\text{ mg cm}^{-2}$). Based on the discharge capacity of the sulfur electrode (600 mAh g^{-1}), the current density is only about 1.3 mAh cm^{-2} . This is a significantly lower value than LIB, which uses a current density of about 4 to 5 mAh cm^{-2} .

Response: Thanks for the reviewer's suggestions, on pages 15-16 in the revised manuscript, we demonstrated that PTMG-HDI-BHDS/LiFSI was capable of supporting higher cathode loading, which was important to meet the requirement for the state-of-the-art lithium batteries. The designed SPE acted as both a Li^+ conductor and binder (experimental section), providing Li^+ pathways in high-loading SPAN cathode (6.8 mg cm^{-2} with a thickness of $\sim 80\text{ }\mu\text{m}$) (Supplementary Fig. 35). The SPAN|PTMG-HDI-BHDS|Li cell delivered a high discharge capacity of 647 mAh g^{-1} with a high areal capacity of 4.4 mAh cm^{-2} and kept stable over 110 cycles (Supplementary Fig. 31). This work provided a promising strategy for the design of high-energy-density solid-state Li-S batteries (see lines 371-376 of page 15, lines 499-500 of page 20, Supplementary Fig. 31 and Supplementary Fig. 35).

Supplementary Fig. 31 Cycling performance of the SPAN|PTMG-HDI-BHDS|Li at 0.3 C with a high loading of 6.8 mg cm⁻².

Supplementary Fig. 35 (a-c) The cross-sectional SEM images of the thicker SPAN cathode. (d) The cross-sectional SEM image of integrated SPAN@SPE. (e, f) the corresponding S mapping and backscattered-electron of integrated SPAN@SPE. (g) The magnifying cross-sectional SEM image of integrated SPAN@SPE. (h, i) the corresponding S mapping and backscattered-electron of integrated SPAN@SPE.

4. **Comment:** To achieve the long cycle life described by the authors, it is

necessary to apply sufficient external pressure to the pouch cell containing the Li metal anode. Please provide an explanation for this requirement and ensure it is adequately reflected in the manuscript.

Response: During the test of pouch cell, it is really necessary to apply pressure to form a better interface contact for the pouch cell, especially for the solid-state batteries. In this experiment, a steel mold was used and the fixed pressure was set at about 200 KPa. Specific details were presented in the experimental section of the revised manuscript (see lines 504-507 of page 20).

5. **Comment:** In this manuscript, authors used too many 'interface impedance' or 'interfacial impedance'. But, in my opinion, interface impedance is not an appropriate expression in electrochemistry. I recommend using the word 'resistance' instead of 'impedance'.

Response: Thanks for the reviewer's suggestions, the word 'impedance' has been replaced by 'resistance' in the revised manuscript (see line 26 of page 2, line 51 of page 3, line 119 of page 6 and line 227 of page 10).

6. **Comment:** The authors suggest that the polymer electrolyte developed by them has a wide potential window and is particularly stable at high potentials, but such a wide potential window is meaningless when using a sulfur cathode. (In this study, the potential is only used up to 3V vs Li/Li⁺). If the authors want to appeal to a wide potential window, add experiments on cathode materials that react at high potentials, otherwise move the related content to the supporting information.

Response: Thanks for the reviewer's suggestions, high-voltage Li-metal batteries (HVLMBs) coupling with solid-state polymer electrolytes (SPEs) have been regarded as a promising strategy to guarantee the high energy density and safety. The target SPE could effectively suppress the decomposition at high oxidation potential (5.1 V), showing advantages in LiNi_{0.8}Co_{0.1}Mn_{0.1}O₂ (NCM811) battery systems. As a result, high-voltage NCM811|PTMG-HDI-BHDS|Li cell

was assembled, which delivered a high discharge capacity of 157.5 mAh g^{-1} with a capacity retention of 80.0% over 200 cycles, effectively suppressing the microstructural degradation and side reactions of NCM811 (see lines 252-254 of page 11 and Supplementary Fig. 17).

Supplementary Fig. 17 (a) voltage profiles and (b) Long-term cycling performance of the NCM811|PTMG-HDI-BHDS|Li cell at 0.5 C.

7. **Comment:** Line 87: What is SPAN? Provide its full name.

Response: Thanks for the reviewer's suggestions, we have checked the manuscript carefully to revise typos. The full name of SPAN (sulfurized polyacrylonitrile) has been added into the revised manuscript (see line 82 of page 4).

8. **Comment:** Line 103: What is polymerized? Is it polymerized? Please check it.

Response: Thank the reviewer for the careful examination, we have checked the manuscript carefully to revise typos (see line 100 of page 5).

9. **Comment:** Apart from the aforementioned errors, numerous typographical errors have been identified. The manuscript contains mistakes and inconsistencies in subscript notation and the usage of symbols. Therefore, it is necessary to edit the manuscript and submit relevant corrections and certificates.

Response: We have checked the manuscript carefully to correct the typos. The subscript notation and symbols have also been carefully checked and corrected. We have polished the language of the manuscript through Springer Nature author services and submitted the corresponding proofs (see line 31 of page 2, line 90 of page 4, line 396 of page 16 and line 451 of page 18).

10. **Comment:** Fig. 3: The axis titles in Fig. 3a are wrong. Please correct it.

Response: Thanks for the reviewer's the careful examination, the axis titles in Fig. 3a have been corrected (see Fig.3a).

Reviewer 3

Reviewer #3 (Remarks to the Author): The paper proposes a cleverly engineered solid polymer electrolyte based on a poly ether-urethane capable

of self-healing. Although the approach proposed is not new in concept (see for example DOI:10.3390/polym15051145 or DOI:10.1007/s40820-023-01075-9), it clearly shows promise in terms of processability and scalability for commercial Li-S applications.

This reviewer has identified a few points worth a more in-depth discussion mainly on some chemical related aspects and on cell processing.

Response: We would like to thank the reviewer's positive comment. We have carefully revised our manuscript according to the reviewer's suggestions. Please see the detailed replies to the following comments:

In the two reviews mentioned by the reviewer, the rational designed self-healing SPEs by using disulfide bonds **is mainly focused on inhibiting the crack or deformation in the electrode materials during the cycling.** However, **the large interfacial ionic transfer barriers between the SPEs (without any solvents) and porous electrodes was despised.** Moreover, after assembling the battery, it is a black box, **there are no effective strategies to track the solid-solid interfacial contact problems in real time.** In our work, the dual-integrated strategy was constructed by skillfully introducing 2-hydroxyethyl disulfide (BHDS) as chain extender. The integrated S@SPE and Li@SPE prepared by the pre-impregnation coating method can solve the multiple interfacial problem of electrode/electrolyte. **The role of disulfide bond with self-healing character mainly focuses on solving the interfaces contact problem between the SPEs on the surface of the two integrated electrodes.** The robust PTMG-HDI-BHDS (88.3 MPa) can restrain the growth of Li dendrites and enable the symmetric cells to deliver a long-term cycling stability for 6000 h. **We further used in situ ultrasonic scanning imaging technique to nondestructively verify the key role of disulfide bonds in repairing the electrolyte interfaces.**

1. **Comment:** Concerning the PTMG oligomer, the authors present results for the average $M_w=2000$. However, from a cursory look at the chemical providers,

it will be very easy to obtain various lengths ranging from very short $M_w=250$ up to longer $M_w=3000$ and beyond. What is the rationale behind the specific polymer length? Can the authors provide data for very short and much longer oligomers? What are the critical effects PTMG length has on the solid electrolyte characteristics?

Response: Thanks for the reviewer's suggestions, the molecular weight of the monomer (PTMG) had a significant effect on the polymerization process. When controlling for the same molar ratio (1:1.2) of monomer PTMG and HDI, the short-chain PTMG250 ($M_w=250 \text{ g mol}^{-1}$) could react violently with HDI at low temperature, the reaction rate was uncontrollable and obvious agglomeration occurred quickly. As another controlled experiment, the reaction rate of PTMG2900 ($M_w=2900 \text{ g mol}^{-1}$) was quite slow even under higher temperature and extended reaction time. The final viscosity of the obtained polymer solution was obviously low, probably due to the poor reactivity of the terminal functional group of the long-chain PTMG2900 when compared with PTMG250. More seriously, the solution of polymerized PTMG2900 was further casted on polytetrafluoroethylene plate, the resulting dried polymer was too sticky to peel off. Therefore, the selection of monomers with appropriate molecular weight is very important for polymerization (see lines 91-94 of page 4 and Supplementary Fig. 1).

Supplementary Fig. 1 Digital photographs of the PTMG-HDI solutions with the (a, b) short-chain PTMG250 and (c, d) long-chain PTMG2900.

2. **Comment:** The polymer electrolyte (including LiFSI) has a relatively low conductivity at RT but an impressively high transference number (0.81). This is a very helpful trait for high-rate capabilities as demonstrated in Fig 5. However, there is nothing immediately apparent justifying such a high transference number as there are no elements that would trap/hinder the FSI⁻ anion, Can the authors explain the origin and mechanism by which such a high value is obtained?

Response: The formation of hydrogen bonds between H atom and F in anions of fluoride salts (LiTFSI and LiFSI) had been demonstrated by NMR in the literature (*Angew. Chem. Int. Ed.* 2021, 60, 10871-10879). Considering that the anions contain a lot of F atoms and the H...F bonds were the strongest hydrogen bonds. The abundant urethane groups in PTMG-HDI-BHDS were able to provide a rich hydrogen bond network with FSI⁻ anions, significantly blocking the free movement of FSI⁻ to a certain extent. The -NH...F hydrogen bonds between urethane groups and FSI⁻ was further confirmed by the upfield displacement of the FSI⁻ peak in the ¹⁹F spectrum (Supplementary Fig. 18).

The abundant ether-oxygen and carbonyl oxygen groups in the structure promoted the dissociation of LiFSI and significantly promoted the free movement of Li⁺. Benefiting from the effect of strong hydrogen bonds and abundant oxygen groups, the number of Li⁺ transference number was significantly increased (see lines 259-263 of page 11, Supplementary Fig. 18 and Ref. 54).

Supplementary Fig. 18 The ¹⁹F NMR spectra of LiFSI in DMSO-*d*₆ without or with presence of PTMG-HDI-BHDS.

3. Comment: The other important aspect of this manuscript is the cell handling, and the fabrication of meaningful 5x6 cm footprint cells is very laudable and an encouraging sign. Many more points appear on this front. The polymer solution appears to be very viscous so one interesting aspect is cathode permeation. Is there a critical thickness (say no more than 20μm) beyond which cathode permeation is failing (the polymer does not reach the bottom layer close to the current collector)? Do the authors have data on thicker cathodes (even if energy density is not a problem as the aim of the manuscript is the S cathode)? Similarly, for full permeation of the standard cathode, is there a critical porosity value? Do the authors have data on porosity as it is not currently treated in version of the work? The situation can be roughly inferred by Fig 6 and Supp

Fig 20 but can the authors provide backscattered electrons imaging as permeation is very hard to judge with the current pictures.

Response: Thanks for the reviewer's suggestions, the solution of SPEs were regulated to the appropriate concentration (70 mg mL^{-1}) with suitable viscosity before casting on the cathode surface.

In the preparation of the cathodes, we had introduced PTMG-HDI-BHDS/LiFSI as a binder and Li^+ conductor into thick SPAN cathode (6.8 mg cm^{-2} with a thickness of $\sim 80 \text{ }\mu\text{m}$) to transport Li^+ (Supplementary Fig. 31) throughout the whole cathodes. The dried cathodes were rolled to minimize the porosity and increase electron conduction, so the porosity was not considered in the work (Supplementary Fig. 35a-c). The cast SPEs solutions were indeed difficult to penetrate into the rolled thick cathodes. However, the transport of Li^+ in the thick cathode does not rely on the small amount of infiltrated polymer solution in the cathode, but rely on the PTMG-HDI-BHDS/LiFSI binder introduced into the cathode in advance. The PTMG-HDI-BHDS/LiFSI polymer membrane on the top of the dual-integrated cathode is designed to reduce the interfacial resistance, exhibit interfacial self-healing ability, and maintain good interfacial contact between the SPE and the electrodes. Therefore, the critical thickness of the cathode is not intensively studied in our work to check whether the polymer reaches the bottom layer close to the current collector.

The continuous electrode/electrolyte interface was formed without any cracks in the S mapping images (Supplementary Fig. 35e and 35h). It was worth noting that the SPE layer and the SPAN layer both contained light elements C, N, O, F, N and S, which were difficult to distinguish in the backscattered-electron SEM images (see line 60 of page 3, lines 346-349 of page 15, line 430 of page 18 and Supplementary Fig. 35).

Supplementary Fig. 35 (a-c) The cross-sectional SEM images of the thicker SPAN cathode. (d) The cross-sectional SEM image of integrated SPAN@SPE. (e, f) the corresponding S mapping and backscattered-electron of integrated SPAN@SPE. (g) The magnifying cross-sectional SEM image of integrated SPAN@SPE. (h, i) the corresponding S mapping and backscattered-electron of integrated SPAN@SPE.

4. **Comment:** Continuing on the polymer solutions, the mixing described in the methods section will yield roughly 3g of electrolyte. Is it possible to mix bigger quantities? What is the stable shelf life of the mixture and are there any special conditions (say fridge storage for example)? How many 5x6cm cells can be coated with a batch of 3g? Did the authors investigate mixing larger batches (say 50g, 100g)?

Response: The reaction condition of the stepwise polymerization is mild, uniform, controllable, and is easily scalable up to the kilogram level. As shown in the Supplementary Fig. 2, PTMG-HDI-BHDS solution with a solid content of 100 g was successfully prepared in a closed glass bottle. Polyurethane with the characteristics of excellent thermal and chemical stability make itself be stored

for a long time. For example, the viscosity and molecular weight of the synthesized polyurethane did not change after one month (Supplementary Fig. 3). However, we recommend that the synthesized PTMG-HDI-BHDS solution should be kept in a dry and sealed atmosphere at room temperature, because the solvent DMAc is relatively easy to absorb water in the air. The total thickness of the coated polymer between the single-layer cathode and Li anode was about 20 μm with a total loading weight of 200 mg. So, 3 g SPE can coat 15 single-layer pouch batteries (see lines 102-103 of page 5 and Supplementary Fig. 2).

Supplementary Fig. 2 Digital photographs of the PTMG-HDI-BHDS solution (1000 mL) prepared by magnifying production.

5. Comment: Lastly, a few points concerning the cells themselves: for obvious reasons, the authors are presenting a very limited set of data concerning a few cells but is there statistical significance in the presented data? How many cells were prepared, and what is the reproducibility ratio? Can the authors provide a figure on statistical significance?

Response: Thanks for the reviewer's suggestions, the results of eight cells (four cells for long-cycling test and the other four cells for rate test) tested under the same condition demonstrate that the cells showed good consistency, and the electrochemical performances were highly reproducible (Supplementary Fig. 30). The good cycling performance attributed to the self-healed multiple

interface contact (see lines 362-363 of page 15 and Supplementary Fig. 30).

Supplementary Fig. 30 Reproducibility of the cell testing. (a) Rate and (b) cycling performance of different SPAN|PTMG-HDI-BHDS|Li cells with a mass loading of 2.0~2.2 mg cm⁻² tested with the same experimental conditions.

6. **Comment:** 6. In comparing Supp Fig 20 and 21, it looks like the electrolyte layer is ~10um thick while in Fig 6c it is more like 20-30um. By what method was the 10um deposited? Is there a critical thickness below which short circuit cannot be avoided and after which energy density is affected at the cell level? What is the authors recommendation on the subject if these cells were to be put into a production line?

Response: The thicknesses of the SPE on integrated SPAN cathode and Li anode were both about ~10 μm. The dual-integrated SPAN|PTMG-HDI-BHDS|Li cells were assembled by fitting the integrated SPAN@SPE with Li@SPE together. Therefore, as shown in Fig. 6c, the total thickness of the electrolyte at the interface after healing was controlled at about ~20 μm.

The integrated SPAN@SPE and Li@SPE was scraped and coated by manual scraper coating method. The coating thickness can be controlled by adjusting the scraping thickness. Specific details have been presented in the experimental section of the received manuscript (see Supplementary Fig. 35).

In order to take into account of the energy density and safety of solid-state batteries, the total thickness of the electrolyte is set to 20 μm, which is similar

to the thickness of the commercial separator. We hold the opinion that too thick SPE film will significantly increase the resistance and reduce the energy density of the battery. Because NCM811 and lithium metal face huge volume expansion during cycling, thin SPE below 20 μm will face the problem of short circuit. In addition, we recommend that more attention should be paid to the places of electrode edges and tabs in assembly line production, which need to be extra blocked to prevent short circuit problems.

7. **Comment:** And speaking of production lines, when using laminated Cu/Li foils for the anode, it is critical to quality check the surface of the foil for any imperfections and impurities. Did the authors take any measure to control the initial Li surface features? What's their experience of the effect of such features on their cells?

Response: In our experience, the oil contamination on the surface of the Cu/Li foils may affect the transport speed of lithium ions, contaminate the electrolyte, and cause side reactions, thereby reducing the capacity and efficiency of the battery. Therefore, we recommend that the tetrahydrofuran solvent is adopted easily remove surface organic contaminants. Besides, the edge of the Cu/Li foils will produce burrs during the cutting process, easily causing battery safety problems, so the surface of the Cu/Li foils needs to be smoother by rolling treatment.

REVIEWER COMMENTS

Reviewer #1 (Remarks to the Author):

The authors responded diligently to the reviewers' questions and revised the manuscript accordingly, conducting additional characterization as requested. The concepts now sound well-explained and coherent. Therefore, I would like to recommend accepting the paper for publication.

Reviewer #3 (Remarks to the Author):

The authors have followed their original publication with an extensive set of additional data. This reviewer commends the work undertaken and sees no reason to further delay publication.

Reviewer #4 (Remarks to the Author):

The manuscript presents the self-healing solid polymer electrolytes with integrated electrode/electrolyte structure for ultralong-life solid-state lithium-sulfur batteries. Interestingly, the introduction of 2-hydroxyethyl disulfide (BHDS) as a chain extender to PTMG-HDI-BHDS forms dynamic covalent disulfide bonds, providing self-healing capability. Additionally, the superior interfacial contact between the SPE and the electrode is attributed to the integrated electrode/electrolyte design. However, the integrated electrode/electrolyte structure fabricated by solvent evaporation method after electrolyte casting is similar to another study. Although the novelty of the work is concerned, the electrochemical performances of Li-S pouch cells and deep investigations are noteworthy. The following issues should be addressed before being accepted.

1. The methodology and purpose of the integrated electrode/electrolyte design in this work are similar to previous study (Angew. Chem. Int. Ed., 2021, 60, 12931-12940). Although the paper is already included in references, it is not mentioned in the experimental section.
2. Typically, enhancing the operating current density would inevitably increase the cell polarizations, leading the charge voltage of the cell to increase while the discharge voltage decreases (e.g. Figure 5b and Supplementary Figure S17). However, Supplementary Fig. 29 exhibits a significant increase in discharge voltage, even though the C-rate increased three times from 0.1 C (1st cycle) to 0.3 C (50, 200, 400, 600, and 700th cycle). Paradoxically, the charge voltage also increases for the same cell. This means that SPAN|PTMG-HDI-BHDS|Li cell shows voltage behavior in which only the charge overpotential increases while the discharge overpotential decreases after the initial cycle. Authors should explain the anomalous voltage behavior of the SPAN|PTMG-HDI-BHDS|Li cell in detail.
3. Based on the wide electrochemical stability window of the PTMG-HDI-BHDS/LiFSI, NCM811|PTMG-HDI-BHDS|Li cell shows 80.0% capacity recovery over 200 cycles even at high voltage. To claim distinction from other SPEs, it is recommended to conduct cell tests using PEO and PTMG-HDI at the same voltage range.
4. Pay attention to unit notation. In Supplementary Fig. 20 and Supplementary Fig. 22, only the unit for areal capacity is written.

Point-by-point response (NCOMMS-23-20631)

Reviewer Comments

Reviewer #1 (Remarks to the Author):

The authors responded diligently to the reviewers' questions and revised the manuscript accordingly, conducting additional characterization as requested. The concepts now sound well-explained and coherent. Therefore, I would like to recommend accepting the paper for publication.

Response: Thanks for the reviewer's recommendation for publication.

Reviewer #3 (Remarks to the Author):

The authors have followed their original publication with an extensive set of additional data.

This reviewer commends the work undertaken and sees no reason to further delay publication.

Response: Thanks for the reviewer's recommendation for publication.

Reviewer #4 (Remarks to the Author):

The manuscript presents the self-healing solid polymer electrolytes with integrated electrode/electrolyte structure for ultralong-life solid-state lithium-sulfur batteries. Interestingly, the introduction of 2-hydroxyethyl disulfide (BHDS) as a chain extender to PTMG-HDI-BHDS forms dynamic covalent disulfide bonds, providing self-healing capability. Additionally, the superior interfacial contact between the SPE and the electrode is attributed to the integrated electrode/electrolyte design. However, the integrated electrode/electrolyte structure fabricated by solvent evaporation method after electrolyte casting is similar to another study. Although the novelty of the work is concerned, the electrochemical performances of Li-S pouch cells and deep investigations are noteworthy. The following issues should be addressed before being accepted.

Response: We would like to thank the reviewer's positive comments. We have carefully revised our manuscript according to the reviewer's suggestions. Please also see the detailed replies to the following comments:

1. The methodology and purpose of the integrated electrode/electrolyte design in this work are similar to previous study (Angew. Chem. Int. Ed., 2021, 60, 12931-12940). Although the paper is already included in references, it is not mentioned in the experimental section.

Response: Thanks for the reviewer's suggestions, the preparation method of the integrated electrode/electrolyte in the literature mentioned by the reviewer provides an important reference value for our work, we further cited in the experimental section (See lines 501-503 of page 20).

2. Typically, enhancing the operating current density would inevitably increase the cell polarizations, leading the charge voltage of the cell to increase while the discharge voltage decreases (e.g. Figure 5b and Supplementary Figure S17). However, Supplementary Fig. 29 exhibits a significant increase in discharge voltage, even though the C-rate increased three times from 0.1 C (1st cycle) to 0.3 C (50, 200, 400, 600, and 700th cycle). Paradoxically, the charge voltage also increases for the same cell. This means that SPAN|PTMG-HDI-BHDS|Li cell shows voltage behavior in which only the charge overpotential increases while the discharge overpotential decreases after the initial cycle. Authors should explain the anomalous voltage behavior of the SPAN|PTMG-HDI-BHDS|Li cell in detail.

Response: Thanks for the reviewer's suggestions, the enhanced current density would inevitably increase the cell polarizations. In our work, for the rate performance of SPAN|PTMG-HDI-BHDS|Li cells, the potential of the first discharge plateau was significant lower compared to the second discharge plateau even with the same current density (0.1 C). The phenomenon of the voltage plateau difference between the first and second discharge curves of

SPAN material is well established in the reported literatures (J. Am. Chem. Soc. 2023,145, 9624–9633; Nano Lett. 2020, 20, 2191–2196; Adv. Funct. Mater. 2019, 29, 1900392). A single voltage plateau of solid phase electrochemical conversion during the first discharge cycle exhibited a large voltage hysteresis, this is owing to the high energy needed to activate the S–C or S–N covalent bonds of SPAN. Then the formation of Li-C-C-Li and Li-C-N-Li bonds can increase the conductivity of activated-SPAN cathode in subsequent cycles and facilitate Li⁺ ions transport, resulting in a lower polarization (*Adv. Funct. Mater.* 2019, 29, 1902929; *Energy Storage Materials* 2022, 50 197–224; *Energy Storage Materials* 2020, 26, 483–493; *Energy Storage Mater.* 2018, 14, 272–278).

Fig. 1 The first two charge/discharge curves of SPAN|PTMG-HDI-BHDS|Li cell at 0.1 C.

3. Based on the wide electrochemical stability window of the PTMG-HDI-BHDS/LiFSI, NCM811|PTMG-HDI-BHDS|Li cell shows 80.0% capacity recovery over 200 cycles even at high voltage. To claim distinction from other SPEs, it is recommended to conduct cell tests using PEO and PTMG-HDI at the same voltage range.

Response: Thanks for the reviewer's suggestions, the cycle performances of NCM811|PEO|Li and NCM811|PTMG-HDI|Li cells, which have been tested earlier, were provided in the revised Supplementary Information. The NCM811|PTMG-HDI|Li cell within a cutoff voltage window of 3.0–4.3 V showed

a significantly lower capacity retention rate, while the capacity of NCM811|PEO|Li decayed rapidly and the battery overcharge occurred after 20 cycles due to poor electrochemical stability (See lines 253-254 and Supplementary Fig. 17b).

Supplementary Fig. 17 (a) voltage profiles of the NCM811|PTMG-HDI-BHDS|Li cell at 0.5 C, (b) Long-term cycling performance of NCM811|SPEs|Li cell at 0.5 C.

4. Pay attention to unit notation. In Supplementary Fig. 20 and Supplementary Fig. 22, only the unit for areal capacity is written.

Response: Thanks for the reviewer's the careful examination, the unit notation of 0.2 mAh cm⁻² in Supplementary Fig. 20 and Supplementary Fig. 22 have been replaced with 0.2 mA cm⁻² (see Supplementary Fig. 20 and Supplementary Fig. 22).

Supplementary Fig. 20 Galvanostatic cycling of Li|PEO|Li pouch cells at the current density of 0.2 mA cm⁻².

Supplementary Fig. 22 Galvanostatic cycling of Li|PTMG-HDI-BHDS|Li pouch cells at the current density of 0.2 mA cm⁻².

REVIEWERS' COMMENTS

Reviewer #4 (Remarks to the Author):

The authors have revised the manuscript by incorporating all the comments. Additionally, they have provided a reasonable and clear response to the question. Hence, I would like to recommend this paper to be published.